# Effective Bayesian Heteroscedastic Regression with Deep Neural Networks

**Alexander Immer**[*,1,2]    **Emanuele Palumbo**[*,1,3]    **Alexander Marx**[†1,3]    **Julia E. Vogt**[†1]

[1]Department of Computer Science, ETH Zurich, Switzerland
[2]Max Planck Institute for Intelligent Systems, Tübingen, Germany
[3]AI Center, ETH Zurich, Switzerland

## Abstract

Flexibly quantifying both irreducible aleatoric and model-dependent epistemic uncertainties plays an important role for complex regression problems. While deep neural networks in principle can provide this flexibility and learn heteroscedastic aleatoric uncertainties through non-linear functions, recent works highlight that maximizing the log likelihood objective parameterized by mean and variance can lead to compromised mean fits since the gradient are scaled by the predictive variance, and propose adjustments in line with this premise. We instead propose to use the natural parametrization of the Gaussian, which has been shown to be more stable for heteroscedastic regression based on non-linear feature maps and Gaussian processes. Further, we emphasize the significance of principled regularization of the network parameters and prediction. We therefore propose an efficient Laplace approximation for heteroscedastic neural networks that allows automatic regularization through empirical Bayes and provides epistemic uncertainties, both of which improve generalization. We showcase on a range of regression problems—including a new heteroscedastic image regression benchmark—that our methods are scalable, improve over previous approaches for heteroscedastic regression, and provide epistemic uncertainty without requiring hyperparameter tuning.

## 1 Introduction

Capturing the *epistemic* (model uncertainty) and *aleatoric* uncertainty (observation noise) allows for computing the predictive variance of a model, which is crucial for areas such as active learning [Houlsby et al., 2011, Kirsch, 2023], reinforcement learning [Osband et al., 2016, Yu et al., 2020] and decision making. Bayesian neural networks allow for modelling both epistemic and aleatoric uncertainty, as such, they lend themselves naturally to this task. Typically, they are applied under the assumption of homoscedasticity, i.e., constant noise [MacKay, 1995, Foong et al., 2019, Kirsch, 2023, and others], but also adaptations of variational inference (VI) to model heteroscedasticity, such as mean-field VI [Graves, 2011], deterministic VI [Wu et al., 2019, DVI], and Monte-Carlo Dropout [Gal and Ghahramani, 2016], have been studied. In this paper, we are interested in learning the epistemic and aleatoric uncertainty in heteroscedastic regression for potentially complex tasks such as image regression, where we are given inputs $\mathbf{x} \in \mathbb{R}^D$ and a scalar response $\mathbf{y} \in \mathbb{R}$, and model $\mathbf{y} \mid \mathbf{x} = \mathbf{x}$ as a conditional Gaussian distribution with mean $\mu(\mathbf{x})$ and standard deviation $\sigma(\mathbf{x})$ being dependent on input $\mathbf{x}$. Inherently, the problem involves robustly modelling the aleatoric uncertainty, corresponding to the variance, which is a relevant problem in econometrics, statistics [Harvey, 1976, Amemiya, 1985], and causal discovery [Guyon et al., 2019, Xu et al., 2022].

---

[*]Equal contribution.   † Shared last author.   Correspondence to: `alexander.immer@inf.ethz.ch`  `alexander.marx@inf.ethz.ch`. Code at `https://github.com/aleximmer/heteroscedastic-nn`.

37th Conference on Neural Information Processing Systems (NeurIPS 2023).

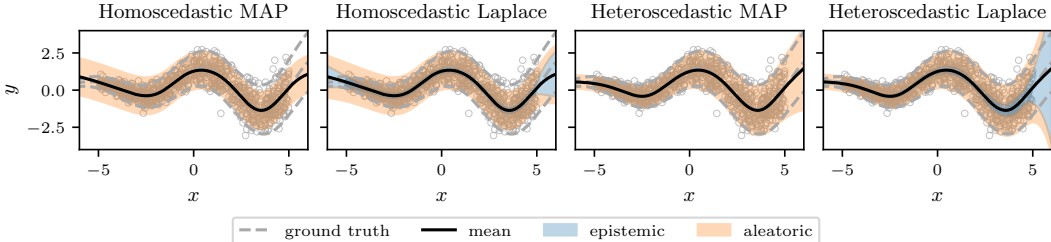

Figure 1: Illustration of the proposed training and posterior predictive of a heteroscedastic Bayesian neural network (right) in comparison to a homoscedastic one (left).

Classical approaches based on linear models and non-linear feature maps [Cawley et al., 2004] attempt to learn $\mu(\mathbf{x})$ and $\sigma(\mathbf{x})$ directly by first fitting the mean function and subsequently fitting the standard deviation from the log-residuals, e.g., feasible generalized least squares (FGLS) [Wooldridge, 2015]. The corresponding objective is, however, only convex if one of the parameters is kept constant, but it is not jointly convex in both parameters [Cawley et al., 2004, Yuan and Wahba, 2004]. To alleviate this problem in the context of Gaussian processes, [Le et al., 2005] propose to reparametrize the objective in terms of the natural parameters, which induces a jointly convex objective. Advantages of the natural parametrization compared to standard FGLS have also been demonstrated for non-linear features maps [Immer et al., 2023a].

In the neural network literature, the standard approach is to model $\mu(\mathbf{x})$ and $\sigma(\mathbf{x})$ as outputs of the network and maximize the corresponding Gaussian log likelihood via stochastic gradient descent [Nix and Weigend, 1994, Lakshminarayanan et al., 2017, Kendall and Gal, 2017], or use two separate neural networks to model mean and standard deviation [Skafte et al., 2019]. As in classical estimators, this parametrization might not be ideal and can lead to overconfident variance estimates [Skafte et al., 2019, Stirn and Knowles, 2020] and possibly compromised mean fits. Two recent strategies aim to adjust for this problem by reducing the influence of the predictive variance on the gradient of the mean. In particular, Seitzer et al. [2022] introduce a surrogate loss, the $\beta-$NLL loss, which regulates the influence of the variance on the gradients of the loss by introducing a stop-gradient operation. As an alternative solution, Stirn et al. [2023] propose architectural constraints coupled with two stop gradient operations, regularizing the heteroscedastic model such that its mean fit is not compromised compared to a homoscedastic baseline. We provide more details to both approaches in Sec. 2.

**Our Contributions**   In comparison to previous work on heteroscedastic regression with neural networks, we take a different perspective postulating that current estimators lack principled regularization. While recent work aims at regularizing the influence of the predictive variance, as we discuss in Sec. 2, we show that this focus can shift the problem to compromising capacity for the variance estimate. Instead, we propose three major modifications to tackle the problem of fitting heteroscedastic neural networks: akin to previous work on GPs [Le et al., 2005] and linear models [Immer et al., 2023a], we propose to re-parameterize the loss using the *natural parametrization* (cf. Sec. 3) which is known to be jointly concave in both parameters. Empirically, we find that this parametrization can be more stable during optimization. Further, we derive an *efficient Laplace approximation to the marginal likelihood* for heteroscedastic regression that can automatically learn regularization via empirical Bayes and provide an early-stopping signal to prevent overfitting without requiring a grid search based on a validation set in Sec. 4.[2] Additionally, the Laplace approximation provides epistemic uncertainty through the *Bayesian posterior predictive*, which generally improves predictive performance. We provide a fast closed-form approximation to the posterior predictive that also provides a simple split into aleatoric and epistemic uncertainties. This predictive is illustrated on the right in Figure 1 and shown in comparison to the prediction with only aleatoric uncertainty (MAP) as well as the corresponding homoscedastic regression solution on the left.

Besides showing that our approach performs favorably on commonly used UCI benchmarks and the CRISPR-Cas13 knockdown efficacy datasets [Stirn et al., 2023], we notice the lack of more complex (heteroscedastic) regression benchmark datasets. To that end, we propose *new image-regression benchmarks* based on image classification datasets. The input images are randomly rotated and the targets are the random rotations with heteroscedastic noise that depends on the label.

---

[2]DVI [Wu et al., 2019] also employs empirical Bayes regularization, but does only apply to MLPs.

## 2 Heteroscedastic Regression with Neural Networks

Due to their universal approximation guarantees [Hornik et al., 1989], deep neural networks have the capacity to solve complex regression problems. As in any regression task, however, unregularized function approximators also have the tendency to overfit to the data. Due to the additional degree of freedom granted by learning both mean and variance in heteroscedastic regression, this problem is amplified, and persists when doing a classical grid search over regularization hyperparameters. In the following, we review prior work that proposes to regularize the influence of the variance and highlight some limitations of these approaches by proposing a new image regression task.

### 2.1 Regularizing the Influence of the Variance

Skafte et al. [2019] note that naively minimizing the Gaussian log likelihood by learning the mean and variance as output of a neural network can lead to overconfident variance estimates that compromise the mean fit. Taking into account the gradients of the negative log likelihood (NLL),

$$\ell_{\mu,\sigma}(\boldsymbol{\theta}) = \tfrac{1}{2}\log\sigma^2(\mathbf{x};\boldsymbol{\theta}) + \frac{(y-\mu(\mathbf{x};\boldsymbol{\theta}))^2}{2\sigma^2(\mathbf{x};\boldsymbol{\theta})} + const, \tag{1}$$

where $\boldsymbol{\theta}$ denote the neural network parameters used to estimate mean and variance, Seitzer et al. [2022] trace the problem to the fact that $\nabla_\mu \ell_{\mu,\sigma}(\boldsymbol{\theta}) = \frac{\mu(\mathbf{x};\boldsymbol{\theta})-y}{\sigma^2(\mathbf{x};\boldsymbol{\theta})}$ heavily depends on the learned variance. To overcome this effect, Seitzer et al. [2022] introduce the $\beta-$NLL loss $\ell_\beta(\boldsymbol{\theta})$, which is equal to $\lfloor\sigma^{2\beta}(\mathbf{x};\boldsymbol{\theta})\rfloor\cdot\ell_{\mu,\sigma}(\boldsymbol{\theta})$, where $\lfloor\cdot\rfloor$ denotes a stop-gradient operation, and $\beta$ is a hyperparameter controlling the dependency of gradients on the predictive variance. As a result, the gradients for $\beta-$NLL are equal to

$$\nabla_\mu \ell_\beta(\boldsymbol{\theta}) = \frac{\mu(\mathbf{x};\boldsymbol{\theta})-y}{\sigma^{2-2\beta}(\mathbf{x};\boldsymbol{\theta})}, \quad \nabla_{\sigma^2}\ell_\beta(\boldsymbol{\theta}) = \frac{\sigma^2(\mathbf{x};\boldsymbol{\theta})-(y-\mu(\mathbf{x};\boldsymbol{\theta}))^2}{2\sigma^{4-2\beta}(\mathbf{x};\boldsymbol{\theta})} . \tag{2}$$

With $\beta = 0$, $\ell_\beta(\boldsymbol{\theta})$ is equivalent to $\ell_{\mu,\sigma}(\boldsymbol{\theta})$, whereas for $\beta = 1$ the gradient with respect to the mean is proportional to the gradient for homoscedastic regression. When setting $0 < \beta \leq 1$ we interpolate between both settings. As an alternative approach, Stirn et al. [2023] propose to decouple the estimation into three networks: a shared representation learner $f_\mathbf{z}$ computes a representation $\mathbf{z}$ from $\mathbf{x}$, which is passed into two individual networks $f_\mu$ and $f_\Sigma$, which receive $\mathbf{z}$ as input and output the mean and covariance matrix, respectively. To ensure that the gradient with respect to the mean is equal to the gradient of the homoscedastic model, they introduce two stop-gradient operations: the first one has an equivalent effect on the mean gradient for $f_\mathbf{z}$ and $f_\mu$ as setting $\beta = 1$ in $\beta$-NLL, and the second one stops any gradient from the variance network $f_\Sigma$ from propergating to $f_\mathbf{z}$ [Stirn et al., 2023]. We provide more details in App. B.

Taking a different perspective, one can view both proposals as implicit regularization techniques for the variance parameter. The surrogate score of Seitzer et al. [2022] regularizes the influence of the predictive variance on the gradient, while the network architecture and stop-gradient operations introduced by Stirn et al. [2023] have a similar, yet stronger regularization effect for the variance—i.e., stopping its influence on the joint representation learner. An additional hurdle presents itself due to the fact that we need to tune the regularization to calibrate the models for a certain dataset, which can have a strong influence on the result, as we show below. Concurrent work [Wong-Toi et al., 2023] also highlights the necessity for regularization in heteroscedastic regression when considering individual mean and variance networks. They find that the networks relative regularization is critical.

### 2.2 Learning Complex Variance Dependencies

Despite alleviating the problem of compromised mean fits, the question arises if such regularization limits the capabilities for learning the aleatoric uncertainty.

In the $\beta$-NLL objective, we need to select the influence of the estimated variance on both gradients, while the effect of scaling is different as can be seen from Equation 2. Due to the typically applied standardization of the data, we expect that $\sigma^2 \leq 1$ and hence the gradient with respect to the variance is amplified stronger compared to the mean. Especially in near-deterministic processes ($\sigma^2 \to 0$) this might be problematic. Further, it is not clear which value of $\beta$ is best suited for a problem a priori introducing another hyperparameter that has to be tuned in addition to tuning regularization hyperparameters. For the approach by Stirn et al. [2023, *Faithful*] the regularization of the variance is

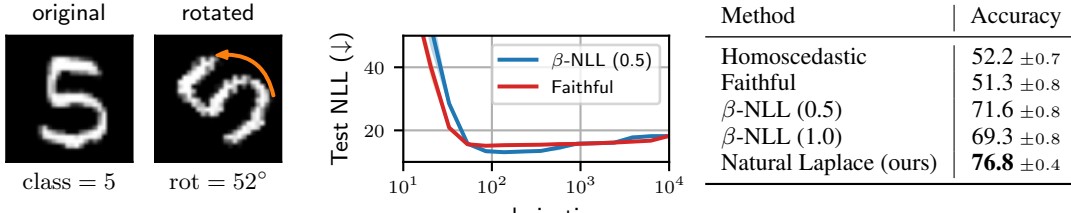

Figure 2: Left: Example heteroscedastic image regression data point with target $y \sim \texttt{rot} + 56\epsilon$. Middle: Test log likelihood for different values of prior precision for $\beta$-NLL and Faithful. Right: Downstream accuracy using a linear model on the last layer representation after learning the heteroscedastic image regression. Compromised variance fit leads to worse downstream accuracy.

more severe due to forcing $\mu(\mathbf{x})$ and $\Sigma(\mathbf{x})$ to share a joint representation that receives no gradient from the variance. If, for example, $\Sigma(\mathbf{x})$ depends on a set of variables that is independent of those influencing the mean, no information about the variance might be contained in the joint representation. To illustrate that only regularizing the variance can be suboptimal, consider the following example.

**Problem 2.1 (Heteroscedastic Regression from Image Data)** *Consider a version of rotated MNIST with rotation angle drawn as $\texttt{rot} \sim Unif(-90, 90)$. We generate the target $y$ as $y = \texttt{rot} + (11c+1)\epsilon$, where $c \in \{0, 1, \ldots, 9\}$ is the image class as integer number and $\epsilon \sim \mathcal{N}(0, 1)$ is an independent noise source. The heteroscedastic regression task of learning the distribution of $y$ given observations (images) $\mathbf{x}$ involves learning a complex non-linear mapping for the variance, while learning the mean only requires learning the rotation angle.*

We train a simple 3-layer MLP of width 500 using the baseline objectives and our approach (*Natural Laplace* introduced later in Secs. 3 and 4) on data generated as described above. First, we note that the test-NLL of $\beta$-NLL and Faithful strongly depends on the regularization strength (cf. Figure 2), which emphasizes the importance of additional regularization. Further, we report how much information about the image classification task is contained in the last layer of the MLP in Figure 2—which provides a proxy on the information about the variance that has been picked up by each approach. The homoscedastic model serves as a control that does not require knowledge about the label. As expected, Faithful achieves a low accuracy on this downstream task since it limits the capability for learning the variance. $\beta$-NLL shows a better performance but is still significantly outperformed by our proposal. The difference becomes even more evident when using a CNN architecture on rotated FashionMNIST as we demonstrate in Sec. 5.3. We also provide a minimal example illustrating the limitations of *Faithful* in App. C, which shows that the bottleneck size of the joint network $f_{\mathbf{z}}$ has a strong influence of the estimation quality of the variance.

As alternatives to Problem 2.1, we consider two generative processes for the rotated image regression task: first, we use a homoscedastic case as ablation with $y = \texttt{rot} + 10\varepsilon$. Further, we use a heteroscedastic noise that is based on the rotational magnitude, $y = \texttt{rot} + \sqrt{|\texttt{rot}|}\varepsilon$. In this case, both mean and variance depend on the same feature, which is a setting where *Faithful* [Stirn et al., 2023] can theoretically work.

## 3 Naturally Parameterized Heteroscedastic Regression

We model a dataset $\mathcal{D} = \{(\mathbf{x}_n, y_n)\}_{n=1}^N$ with $N$ pairs of input $\mathbf{x}_n \in \mathbb{R}^D$ and scalar response $y_n \in \mathbb{R}$ using the natural form of the Gaussian likelihood with unknown variance, as first introduced for the case of Gaussian processes [Le et al., 2005] and recently applied in the context of causal discovery with maximum likelihood [Immer et al., 2023a]. The relationship between the natural parameters $\boldsymbol{\eta}$ and the mean $\mu$ and variance $\sigma^2$ of the parametrization is

$$\eta_1 = \tfrac{\mu}{\sigma^2} \text{ and } \eta_2 = -\tfrac{1}{2\sigma^2} < 0, \tag{3}$$

which can be understood as the signal-to-variance ratio and the negative precision (inverse variance). We model these natural parameters using a neural network $\mathbf{f}(\mathbf{x}; \boldsymbol{\theta}) \in \mathbb{R}^2$ that is parameterized by weights $\boldsymbol{\theta} \in \mathbb{R}^P$ and acts on the inputs $\mathbf{x} \in \mathbb{R}^D$. To satisfy the constraint that $\eta_2 < 0$, we *link* $\mathbf{f}$ to $\boldsymbol{\eta}$ using a positive function $g_+ : \mathbb{R} \mapsto \mathbb{R}_+$ and have the following mapping:

$$\eta_1(\mathbf{x}; \boldsymbol{\theta}) = f_1(\mathbf{x}; \boldsymbol{\theta}) \text{ and } \eta_2(\mathbf{x}; \boldsymbol{\theta}) = -g_+(f_2(\mathbf{x}; \boldsymbol{\theta})). \tag{4}$$

We use either the exponential $g_+(\cdot) = \frac{1}{2}\exp(\cdot)$ or softplus $g_+(\cdot) = \frac{1}{\beta}\log(1 + \exp(\cdot))$ as typical for heteroscedastic regression with mean-variance parametrization. Mathematically, the *heteroscedastic Gaussian log likelihood of our model* is given by

$$\log p(y|\mathbf{x}, \boldsymbol{\theta}) = \begin{bmatrix} \eta_1(\mathbf{x};\boldsymbol{\theta}) \\ \eta_2(\mathbf{x};\boldsymbol{\theta}) \end{bmatrix}^\top \begin{bmatrix} y \\ y^2 \end{bmatrix} + \frac{\eta_1(\mathbf{x};\boldsymbol{\theta})^2}{4\eta_2(\mathbf{x};\boldsymbol{\theta})} + \frac{1}{2}\log(-2\eta_2(\mathbf{x};\boldsymbol{\theta})) + const, \qquad (5)$$

which we also denote by $-\ell_{\boldsymbol{\eta}}(\boldsymbol{\theta})$. Assuming the data are *i.i.d.*, we have $\log p(\mathcal{D}|\boldsymbol{\theta}) = \sum_{n=1}^N \log p(y_n|\mathbf{x}_n, \boldsymbol{\theta})$. Immer et al. [2023a] used this maximum likelihood objective for bivariate causal discovery using linear models and small neural networks. Further, the gradient and Hessian take on simple forms due to the properties of the natural parametrization.

### 3.1 Gradients of the Natural parametrization

Similar to Seitzer et al. [2022] and Stirn et al. [2023], we also inspect the gradients of the corresponding negative log likelihood with respect to $\boldsymbol{\eta}$, leading to

$$\nabla_{\eta_1}\ell_{\boldsymbol{\eta}}(\boldsymbol{\theta}) = -\frac{\eta_1(\mathbf{x};\boldsymbol{\theta})}{2\eta_2(\mathbf{x};\boldsymbol{\theta})} - y, \quad \nabla_{\eta_2}\ell_{\boldsymbol{\eta}}(\boldsymbol{\theta}) = \frac{(\eta_1(\mathbf{x};\boldsymbol{\theta}))^2}{4(\eta_2(\mathbf{x};\boldsymbol{\theta}))^2} - \frac{1}{2\eta_2(\mathbf{x};\boldsymbol{\theta})} - y^2 . \qquad (6)$$

The gradients by themselves cannot be directly linked to mean and variance updates. We note, however, that if we relate the natural parameters to mean and variance, i.e., we compute $\mu(\mathbf{x};\boldsymbol{\theta})$ as $-\frac{\eta_1(\mathbf{x};\boldsymbol{\theta})}{2\eta_2(\mathbf{x};\boldsymbol{\theta})}$ and $\sigma^2(\mathbf{x};\boldsymbol{\theta})$ as $-\frac{1}{2\eta_2(\mathbf{x};\boldsymbol{\theta})}$, then $\nabla_{\eta_1}\ell_{\boldsymbol{\eta}}(\boldsymbol{\theta})$ reduces to $\mu(\mathbf{x};\boldsymbol{\theta}) - y$, and similarly, $\nabla_{\eta_2}\ell_{\boldsymbol{\eta}}(\boldsymbol{\theta})$ to $\sigma^2(\mathbf{x};\boldsymbol{\theta}) - (y^2 - (\mu(\mathbf{x};\boldsymbol{\theta}))^2)$, which would be desired because these are simply separate residuals for mean and variance [Seitzer et al., 2022, Stirn et al., 2023]. Empirically, we observe that this parametrization can be more stable to train and less prone to insufficient regularization as also observed previously for Gaussian process and ridge regression [Le et al., 2005, Immer et al., 2023a].

### 3.2 Regularization using Bayesian Inference

Due to the expressiveness of heteroscedastic regression, regularization is crucial to obtain well-generalizing models (cf. Figure 2). We achieve regularization in two ways: first, we regularize parameters towards low norm using classical $\ell^2$ regularization, which corresponds to a Gaussian prior on the parameters. Further, we use a Bayesian posterior predictive, which additionally accounts for uncertainties of the model as depicted in the illustration (Figure 1, right).

For effective regularization of deep neural networks, we use a layer-wise Gaussian prior on the parameters given by $p(\boldsymbol{\theta}|\boldsymbol{\delta}) = \prod_l \mathcal{N}(\boldsymbol{\theta}_l; \mathbf{0}, \delta_l^{-1}\mathbf{I})$. In the case of the last layer or a simple linear model with the natural heteroscedastic likelihood, it induces a mode on $\eta_1(\mathbf{x}) = 0$ and $\eta_2(\mathbf{x}) = -\frac{1}{2}$, which corresponds to zero mean and unit variance and is reasonable for standardized response variables. The layer-wise prior further allows to differently regularize parts of the neural network and has been observed to improve generalization in image classification [Immer et al., 2021a, Daxberger et al., 2021, Antorán et al., 2022]. Concurrent work by also suggests that this might be the case for heteroscedastic regression [Wong-Toi et al., 2023]. However, optimizing a regularization parameter per layer is intractable using a validation-based grid search.

To optimize layer-wise prior precisions and obtain a posterior predictive, we make use of Bayesian inference. Combining the natural Gaussian likelihood $p(\mathcal{D}|\boldsymbol{\theta})$ (below Equation 5) with the prior $p(\boldsymbol{\theta}|\boldsymbol{\delta})$, we have the joint distribution $p(\mathcal{D}, \boldsymbol{\theta}|\boldsymbol{\delta})$, which corresponds to a regularized objective. According to Bayes' theorem, we have the posterior $p(\boldsymbol{\theta}|\mathcal{D}, \boldsymbol{\delta}) \propto p(\mathcal{D}, \boldsymbol{\theta}|\boldsymbol{\delta})$. The normalization constant, also referred to as *marginal likelihood*, is given by $p(\mathcal{D}|\boldsymbol{\delta}) = \int p(\mathcal{D}, \boldsymbol{\theta}|\boldsymbol{\delta})\mathrm{d}\boldsymbol{\theta}$ and gives us the Type II maximum likelihood objective to optimize the prior precisions $\boldsymbol{\delta}$. This procedure is referred to as *empirical Bayes* (EB). Inferring the posterior distribution of the neural network parameters, we further have access to the posterior predictive $p(y_*|\mathbf{x}_*, \mathcal{D})$ for a new data point $\mathbf{x}_*$. By averaging over multiple hypotheses from the posterior, the predictive can be better regularized than a single model [Wilson, 2020]. Unfortunately, inference is intractable for deep neural networks.

## 4 Approximate Inference with a Laplace Approximation

We develop a scalable Laplace approximation for the posterior in heteroscedastic regression with deep neural networks. The Laplace approximation [MacKay, 1995] is an effective method for

approximating the marginal likelihood, posterior, and predictive in deep learning [Daxberger et al., 2021]. In comparison to other approximate inference methods, it can rely on effective training algorithms developed for deep learning and also offers a differentiable marginal likelihood estimate that enables empirical Bayes (EB). Efficient curvature approximations further make it scalable to deep learning [Ritter et al., 2018]. We extend these to the heteroscedastic regression setting.

Laplace approximates the posterior locally at a mode of the posterior with a Gaussian distribution, $p(\boldsymbol{\theta}|\mathcal{D}, \boldsymbol{\delta}) \approx \mathcal{N}(\boldsymbol{\theta}; \boldsymbol{\theta}_*, \boldsymbol{\Sigma})$. The mean is given by a stationary point of the posterior, $\boldsymbol{\theta}_* = \arg\max_{\boldsymbol{\theta}} \log p(\mathcal{D}, \boldsymbol{\theta}|\boldsymbol{\delta})$, and the covariance by the curvature at that mode, $\boldsymbol{\Sigma}^{-1} = \nabla^2_{\boldsymbol{\theta}} \log p(\mathcal{D}, \boldsymbol{\theta}|\boldsymbol{\delta})|_{\boldsymbol{\theta}=\boldsymbol{\theta}_*}$. This is due to a second-order Taylor approximation of the log posterior around the mode. The mean is hence the result of neural network training, however, the covariance requires estimating and inverting a Hessian, which is typically intractable.

## 4.1 Linearized Laplace for Natural Heteroscedastic Regression

The linearized Laplace approximation [MacKay, 1995, Khan et al., 2019, Foong et al., 2019, Immer et al., 2021b] overcomes issues of the vanilla Laplace approximation. Linearizing the neural network about the parameters at the mode, the Hessian, which is the generalized Gauss-Newton in this case [Martens, 2020], becomes positive semidefinite and offers efficient structured approximations. In particular, we have the following Hessian approximation due to linearization

$$\boldsymbol{\Sigma}^{-1} \approx \sum_{n=1}^{N} \mathbf{J}_*(\mathbf{x}_n)^{\mathsf{T}}[-\nabla^2_{\boldsymbol{\eta}} \log p(y_n|\mathbf{x}_n, \boldsymbol{\theta}_*)]\mathbf{J}_*(\mathbf{x}_n) + \nabla^2_{\boldsymbol{\theta}} \log p(\boldsymbol{\theta}|\boldsymbol{\delta})|_{\boldsymbol{\theta}=\boldsymbol{\theta}_*}, \qquad (7)$$

where $[\mathbf{J}_*(\mathbf{x})]_{cp} = \frac{\partial \eta_c(\mathbf{x};\boldsymbol{\theta})}{\partial \theta_p}|_{\boldsymbol{\theta}=\boldsymbol{\theta}_*}$ is the Jacobian of the neural network $\boldsymbol{\eta}(\mathbf{x}; \boldsymbol{\theta})$ at the mode. The first summand is the generalized Gauss-Newton and the second term is the Hessian of the prior, which is simply a diagonal matrix constructed from entries $\delta_l > 0$, the layer-wise regularization parameters.

Due to the natural parametrization of the likelihood, the Hessian approximation of the linearized Laplace approximation is guaranteed to be positive definite. Since the log prior Hessian is diagonal with entries $\delta_l > 0$, one only has to show that the negative log likelihood Hessian, $-\nabla^2_{\boldsymbol{\eta}} \log p(y|\mathbf{x}, \boldsymbol{\theta}_*)$, is positive semidefinite, which is simply a property of naturally parameterized exponential families [Martens, 2020]. Note that this would not be the case for the mean-variance parametrization[3]. For efficient computation, we can decompose the log likelihood Hessian as

$$\boldsymbol{\Lambda}_*(\mathbf{x}) \stackrel{\text{def}}{=} \nabla^2_{\boldsymbol{\eta}} \log p(y|\mathbf{x}, \boldsymbol{\theta}_*) = \begin{bmatrix} \frac{1}{2\eta_2(x;\boldsymbol{\theta}_*)} & -\frac{\eta_1(x;\boldsymbol{\theta}_*)}{2\eta_2(x;\boldsymbol{\theta}_*)^2} \\ -\frac{\eta_1(x;\boldsymbol{\theta}_*)}{2\eta_2(x;\boldsymbol{\theta}_*)^2} & \frac{\eta_1(x;\boldsymbol{\theta}_*)^2}{2\eta_2(x;\boldsymbol{\theta}_*)^3} - \frac{1}{2\eta_2(x;\boldsymbol{\theta}_*)^2} \end{bmatrix} \qquad (8)$$

$$= \begin{bmatrix} -\frac{1}{\sqrt{-2\eta_2(x;\boldsymbol{\theta}_*)}} & \frac{\eta_1(x;\boldsymbol{\theta}_*)}{\eta_2(x;\boldsymbol{\theta}_*)\sqrt{-2\eta_2(x;\boldsymbol{\theta}_*)}} \end{bmatrix}^2 - \begin{bmatrix} 0 & \frac{1}{\sqrt{2}\eta_2(x;\boldsymbol{\theta}_*)} \end{bmatrix}^2 \qquad (9)$$

$$\stackrel{\text{def}}{=} \lambda_1(\mathbf{x})\lambda_1(\mathbf{x})^{\mathsf{T}} - \lambda_2(\mathbf{x})\lambda_2(\mathbf{x})^{\mathsf{T}}, \qquad (10)$$

where the square indicates the outer product. This alleviates the need to compute this matrix and instead allows to work with outer products of Jacobian-vector products to compute the covariance in Equation 7. However, the full Hessian approximation remains quadratic in the number of neural network weights. We tackle this issue with a scalable Kronecker-factored approximation.

## 4.2 Scalable Kronecker-Factored Hessian Approximation

To enable the application of the Laplace approximation to heteroscedastic deep neural networks and large datasets, we use a layer-wise Kronecker-factored approximation [KFAC; Martens and Grosse, 2015, Botev et al., 2017]. To overcome the quadratic scaling in the number of parameters, KFAC makes two efficient approximations: first, it constructs a block-diagonal approximation to $\mathbf{H}$ from blocks $\mathbf{H}_l$ per layer $l$. Second, each block $\mathbf{H}_l$ is approximated as a Kronecker product that enables efficient storage and computation using only the individual factors. In the following, we revise KFAC for linear layers and define it for the specific case of the heteroscedastic natural Gaussian likelihood. The same derivation applies similarly to other layer types [Osawa, 2021].

Using the Hessian decomposition of the natural log likelihood in Equation 10, we derive a KFAC approximation that can efficiently be computed in a closed-form. We can write the Jacobian of a

---

[3]In Sec. 4.5 we show how Laplace can be applied for the mean-variance parametrization despite that.

fully connected layer that maps a $D$ to a $D'$-dimensional representation as a Kronecker product $\mathbf{J}_l(\mathbf{x}_n)^\mathsf{T} = \mathbf{a}_{l,n} \otimes \mathbf{g}_{l,n}$ with $\mathbf{a}_{l,n} \in \mathbb{R}^{D \times 1}$ as the layer's input and $\mathbf{g}_{l,n} \in \mathbb{R}^{D' \times 2}$ as transposed Jacobian w.r.t. the output, both for the input $\mathbf{x}_n$. Following Martens and Grosse [2015] and Botev et al. [2017], we then have the KFAC approximation

$$[\boldsymbol{\Sigma}^{-1}]_l = \sum_{n=1}^N [\mathbf{a}_{l,n} \otimes \mathbf{g}_{l,n}] \boldsymbol{\Lambda}_n [\mathbf{a}_{l,n} \otimes \mathbf{g}_{l,n}]^\mathsf{T} + \delta_l \mathbf{I} = \sum_{n=1}^N [\mathbf{a}_{l,n}\mathbf{a}_{l,n}^\mathsf{T}] \otimes [\mathbf{g}_{l,n}\boldsymbol{\Lambda}_n\mathbf{g}_{l,n}^\mathsf{T}] + \delta_l \mathbf{I}$$

$$\approx \frac{1}{N}\left[\sum_{n=1}^N \mathbf{a}_{l,n}\mathbf{a}_{l,n}^\mathsf{T}\right] \otimes \left[\sum_{n=1}^N \sum_{k=1}^2 \mathbf{g}_{l,n}\boldsymbol{\lambda}_{n,k}\boldsymbol{\lambda}_{n,k}^\mathsf{T}\mathbf{g}_{l,n}^\mathsf{T}\right] + \delta_l \mathbf{I} \overset{\text{def}}{=} \mathbf{A}_l \otimes \mathbf{B}_l + \delta_l \mathbf{I}, \tag{11}$$

where $\boldsymbol{\lambda}_{n,k} \in \mathbb{R}^{2 \times 1}$ is due to to the decomposition of $\boldsymbol{\Lambda}_n$ into outer products Equation 10 and the approximation is due to exchanging the sum and product. Conveniently, the terms $\mathbf{g}_{l,n}\boldsymbol{\lambda}_{n,k}$ can be computed efficiently using two Jacobian-vector products and the Kronecker factors can then be extracted within the second-order framework of Osawa [2021]. To the best of our knowledge, this is the first instantiation of KFAC for heteroscedastic regression. While we derive it for the Laplace approximation, it could also be useful for optimization [Martens and Grosse, 2015].

### 4.3 Empirical Bayes for Automatic Regularization

To automatically regularize the heteroscedastic neural network, we use an empirical Bayes (EB) procedure that optimizes the layer-wise prior precisions, $\delta_l$, during training by maximizing the Laplace approximation to the *marginal likelihood* [Immer et al., 2021a]. This procedure can exhibit a Bayesian variant of Occam's razor [Rasmussen and Ghahramani, 2000] and trades off model fit and complexity. Although online training violates the stationarity assumption of the Laplace approximation, it has been observed to work well in practice [Immer et al., 2021a, 2023b, Daxberger et al., 2021, Lin et al., 2023]. We use gradient-based optimization of the log marginal likelihood,

$$\log p(\mathcal{D}|\boldsymbol{\delta}) \approx \log p(\mathcal{D}|\boldsymbol{\theta}_*) + \log p(\boldsymbol{\theta}_*|\boldsymbol{\delta}) + \tfrac{1}{2}\log|\boldsymbol{\Sigma}| + \tfrac{P}{2}\log 2\pi \propto \log p(\boldsymbol{\theta}_*|\boldsymbol{\delta}) + \tfrac{1}{2}\log|\boldsymbol{\Sigma}|, \tag{12}$$

which crucially requires differentiating the log-determinant w.r.t. $\boldsymbol{\delta}$, which is only tractable for small neural networks. For deep neural networks, it can be done efficiently using the KFAC approximation derived in Sec. 4.2 by eigendecomposition of the individual Kronecker factors [Immer et al., 2021a, 2022]. In practice, we compute the marginal likelihood approximation every few epochs to adapt the regularization, which effectively mitigates overfitting, and use it as an early-stopping criterion. The detailed empirical Bayes training algorithm is also described in Alg. 1.

### 4.4 Posterior Predictive for Epistemic Uncertainties

We use the linearized posterior predictive that is in line with the linearized Laplace posterior approximation and performs typically better than sampling weights directly [Immer et al., 2021b]. We define the linearized neural network as $\boldsymbol{\eta}_*^{\text{lin}}(\mathbf{x}; \boldsymbol{\theta}) \overset{\text{def}}{=} \boldsymbol{\eta}(\mathbf{x}; \boldsymbol{\theta}_*) + \mathbf{J}_*(\mathbf{x})(\boldsymbol{\theta} - \boldsymbol{\theta}_*)$. Due to the Gaussianity of the Laplace approximation, $\mathcal{N}(\boldsymbol{\theta}_*; \boldsymbol{\Sigma})$, and the linearization, we can express the function-space posterior as a Gaussian on the natural parameters $\boldsymbol{\eta}_*^{\text{lin}}(\mathbf{x}) \sim \mathcal{N}(\boldsymbol{\eta}(\mathbf{x}; \boldsymbol{\theta}_*); \mathbf{J}_*(\mathbf{x})\boldsymbol{\Sigma}\mathbf{J}_*(\mathbf{x})^\mathsf{T}) \overset{\text{def}}{=} q(\boldsymbol{\eta}|\mathbf{x})$. A straightforward way to approximate the posterior predictive is then a Monte-Carlo estimate of $p(y_*|\mathbf{x}_*, \mathcal{D}) \approx \int p(y_*|\mathbf{x}_*, \boldsymbol{\eta})q(\boldsymbol{\eta}|\mathbf{x}_*)\,\mathrm{d}\boldsymbol{\eta}$ by sampling multiple $\boldsymbol{\eta}_*^{\text{lin}}(\mathbf{x})$.

Alternatively, we propose to use an approximation to the posterior predictive that can be computed in a closed-form without sampling, similar to the *probit approximation* commonly used in the classification setting [Daxberger et al., 2021]. To enable a closed-form posterior predictive, we restrict ourselves to the epistemic uncertainty about the mean as proposed by Le et al. [2005] for heteroscedastic Gaussian process regression. Instead of linearizing the natural parameters, we first transform them to the mean and variance using the inverse mapping of Equation 3, i.e., we have $\mu(\mathbf{x}; \boldsymbol{\theta}) = -\frac{\eta_1(\mathbf{x}; \boldsymbol{\theta})}{2\eta_2(\mathbf{x}; \boldsymbol{\theta})}$ and $\sigma^2(\mathbf{x}; \boldsymbol{\theta}_*) = -\frac{1}{2\eta_2(\mathbf{x}; \boldsymbol{\theta}_*)}$. Next, we only linearize the mean function $\mu_*^{\text{lin}}(\mathbf{x}; \boldsymbol{\theta}) = \mu(\mathbf{x}; \boldsymbol{\theta}_*) + \mathbf{J}_{*,\mu}(\mathbf{x})(\boldsymbol{\theta} - \boldsymbol{\theta}_*)$, where $\mathbf{J}_{*,\mu}(\mathbf{x})$ is the Jacobian of the mean, and have

$$p(y_*|\mathbf{x}_*, \mathcal{D}) \approx \int \mathcal{N}(y_*|\mu_*^{\text{lin}}(\mathbf{x}_*), \sigma^2(\mathbf{x}; \boldsymbol{\theta}_*))\mathcal{N}(\boldsymbol{\theta}; \boldsymbol{\theta}_*, \boldsymbol{\Sigma})\,\mathrm{d}\boldsymbol{\theta}$$

$$= \mathcal{N}(y_*; \mu(\mathbf{x}; \boldsymbol{\theta}_*), \underbrace{\mathbf{J}_{*,\mu}(\mathbf{x}_*)\boldsymbol{\Sigma}\mathbf{J}_{*,\mu}(\mathbf{x}_*)^\mathsf{T}}_{\text{epistemic}} + \underbrace{\sigma^2(\mathbf{x}; \boldsymbol{\theta}_*)}_{\text{aleatoric}}), \tag{13}$$

where the epistemic and aleatoric uncertainty about the mean are clearly split. An example of this posterior predictive approximation is shown in Figure 1.

### 4.5 Laplace Approximation for Mean-Variance parametrization

Using the natural parameter mapping, it is possible to apply above Laplace approximations to the mean-variance parametrization and profit from the empirical Bayes and posterior predictive procedures. However, because the negative log likelihood Hessian w.r.t. the mean and variance parameters can be indefinite, this does not work naively. By mapping $\mu, \sigma^2$ to the natural parameters using Equation 3, all above derivations apply and correspond to a separate Gauss-Newton approximation of the log likelihood with Jacobians of the mapping and the natural log likelihood Hessian (see App. A).

### 4.6 Limitations

As common for heteroscedastic regression, we make the assumption that the conditional distribution of the target given the observations follows a Gaussian distribution. A practitioner should keep in mind that the full Laplace approximation has high computational complexity, and should resort to the KFAC approximation, which we also use for more complex settings in experiments. Further, the true posterior of neural networks is in most cases multimodal while our Laplace approximation only covers a single mode. However in practical settings, the Laplace approximation provides strong results [Daxberger et al., 2021] and can even be ensembled [Eschenhagen et al., 2021].

## 5 Experiments

We evaluate the effectiveness of the natural parameterization compared to the mean-variance (naive) one, and empirical Bayes (EB) to optimizing a single regularization parameter using a grid search on the validation set (GS), and the MAP prediction vs a Bayesian posterior predictive (PP) in comparison to state-of-the-art baselines on three experimental settings: the UCI regression benchmark [Hernandez-Lobato and Adams, 2015], which is also well-established for heteroscedastic regression [Seitzer et al., 2022, Stirn et al., 2023], the recently introduced CRISPR-Cas13 gene expression datasets [Stirn et al., 2023], and our proposed heteroscedastic image-regression dataset (cf. Problem 2.1) in three noise variants. For UCI regression, we use the full Laplace approximation and we resort to the KFAC approximation for the remaining tasks for computational efficiency. Further, we use the proposed closed-form posterior predictive for our Laplace approximations. If applicable, as for our methods, we test the effect of having a Bayesian posterior predictive (PP) in comparison to the point estimate.

**Baselines** As control to assess the benefit of heteroscedastic aleatoric uncertainty, we include a homoscedastic model with EB and Laplace posterior predictive. In addition, we include the mean-variance parameterization of the negative log likelihood loss for heteroscedastic regression (*Naive NLL*). As competitive baselines, we include the recently proposed $\beta$-*NLL* [Seitzer et al., 2022] and *Faithful* [Stirn et al., 2023] heteroscedastic losses. Finally we include well-established approaches for heteroscedastic regression with Bayesian neural networks, namely mean-field variational inference (*VI*) [Graves, 2011] and Monte-Carlo Dropout [Gal and Ghahramani, 2016] (*MC-Dropout*). Note that for our VI baseline we employ Flipout [Wen et al., 2018] to improve gradient estimation.

Details on model architectures, hyperparameter tuning, and additional results are in App. D.

### 5.1 UCI Regression

In Table 1 we report results on the UCI regression datasets [Hernandez-Lobato and Adams, 2015] for the compared models in terms of test log likelihood. We underline the best performance for each dataset, and bold results which are not statistically distinguishable from best performance. That is, if the mean performance of a model falls within the region of the mean plus minus two standard errors of the best performing model. The two rightmost columns report the total number of *wins* and *ties* for each model: wins are the number of datasets in which the given model achieves the best performance, while ties are the number of datasets in which the model achieves results which are not statistically distinguishable from the best performing model. The results validate the effectiveness of EB regularization on different heteroscedastic loss parameterizations. Notably, with the Bayesian posterior predictive (PP) our proposed methods considerably improve in performance in comparison to using a point estimate, and significantly outperform existing state-of-the-art approaches. Finally the results confirm that, particularly when the point prective is used, using the Natural parameterization of the heteroscedastic NLL improves training stability.

| Objective | Regular- ization | Posterior Predictive | LL (↑) boston | concrete | energy | kin8nm | naval | plant | wine | yacht | Wins | Ties |
|---|---|---|---|---|---|---|---|---|---|---|---|---|
| Homoscedastic | EB | ✓ | -2.51 (0.04) | -3.03 (0.03) | -0.71 (0.04) | 1.29 (0.01) | 5.58 (0.45) | **-2.83 (0.02)** | **-0.94 (0.02)** | -1.05 (0.14) | 0 | 2 |
| Naive NLL | GS | ✗ | -2.88 (0.2) | -3.96 (0.32) | -1.06 (0.16) | 1.31 (0.01) | 3.88 (0.27) | -2.85 (0.04) | **-0.95 (0.02)** | -0.60 (0.18) | 0 | 1 |
| $\beta$-NLL (0.5) | GS | ✗ | -2.61 (0.06) | -3.45 (0.24) | -0.96 (0.13) | 1.30 (0.01) | 2.60 (2.70) | -2.85 (0.04) | **-0.94 (0.02)** | -1.39 (0.49) | 0 | 1 |
| $\beta$-NLL (1.0) | GS | ✗ | -2.71 (0.06) | -3.13 (0.04) | -0.88 (0.07) | 1.29 (0.01) | 6.47 (0.12) | -2.86 (0.03) | **-0.94 (0.02)** | -1.09 (0.20) | 0 | 1 |
| Faithful | GS | ✗ | -2.68 (0.07) | -3.12 (0.06) | -1.10 (0.11) | 1.29 (0.01) | 6.41 (0.14) | **-2.80 (0.02)** | -0.95 (0.02) | -1.32 (0.31) | 0 | 2 |
| MC-Dropout | GS | ✗ | -3.01 (0.24) | -3.05 (0.03) | -1.61 (0.03) | 1.17 (0.01) | 5.64 (0.03) | **-2.82 (0.01)** | -1.12 (0.07) | -1.01 (0.08) | 0 | 1 |
| VI | GS | ✗ | -2.62 (0.08) | -3.10 (0.04) | -1.55 (0.03) | 1.31 (0.01) | 5.70 (0.24) | **-2.82 (0.02)** | -1.07 (0.07) | -1.16 (0.05) | 0 | 1 |
| Naive NLL | EB | ✓ | -2.47 (0.04) | **-2.88 (0.03)** | **-0.46 (0.03)** | 1.37 (0.01) | 6.38 (0.16) | **-2.79 (0.02)** | **-0.93 (0.02)** | **-0.05 (0.15)** | 5 | 6 |
|  |  | ✗ | -6.03 (1.82) | -3.62 (0.26) | -0.97 (0.15) | **1.36 (0.01)** | -10.9 (8.01) | -2.83 (0.04) | -2.26 (0.51) | -1.27 (0.51) | 0 | 2 |
| Natural NLL | GS | ✓ | -2.46 (0.04) | **-2.92 (0.04)** | -0.74 (0.06) | 1.32 (0.01) | 6.66 (0.01) | **-2.76 (0.02)** | **-0.94 (0.02)** | -0.29 (0.07) | 0 | 4 |
|  |  | ✗ | -2.57 (0.08) | -2.95 (0.05) | -0.69 (0.08) | 1.31 (0.01) | 6.69 (0.01) | **-2.76 (0.02)** | **-0.95 (0.02)** | -0.87 (0.28) | 1 | 3 |
|  | EB | ✓ | -2.36 (0.03) | **-2.93 (0.02)** | -0.71 (0.03) | **1.36 (0.01)** | 6.66 (0.01) | -2.76 (0.02) | **-0.94 (0.02)** | -0.51 (0.04) | 2 | 5 |
|  |  | ✗ | -2.51 (0.09) | -2.99 (0.04) | -0.65 (0.05) | **1.35 (0.01)** | **6.68 (0.01)** | **-2.77 (0.02)** | -1.00 (0.03) | -0.60 (0.08) | 0 | 3 |

Table 1: Test log likelihood results for the compared models on the UCI regression datasets, where we report mean and standard error (dataset names in italic). Underlining of results indicates best performance while bolded results are statistically identical to best performance. The bottom half shows the proposed methods, which have the best performance overall.

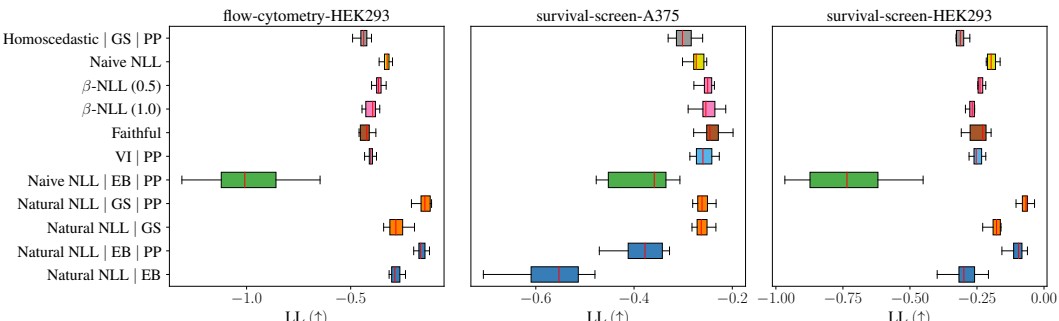

Figure 3: Box-plots reporting test log likelihood results on the three CRISPR datasets, that vary with respect to cell and screen type. The natural parametrization, and using the posterior predictive improve the performance. Note that NLL mean-variance (MAP) and MC-Dropout markedly underperform in this context and are hence excluded from the figure.

## 5.2 CRISPR Datasets for Gene Knockdown Efficacy

As Stirn et al. [2023] highlight, heteroscedasticity is an often encountered problem in natural sciences. To test heteroscedastic regression approches in such a setting, they introduce a set of datasets to model the efficacy of the CRISPR-Cas13 system for gene knockdown. In particular, Cas13 is a system that targets specific RNA transcripts to temporarily decrease gene expression. Decreasing gene expression levels serves multiple purposes in molecular biology, including the understanding of the function of specific genes and for therapeutic aims. In particular, the authors propose three datasets, and for each dataset multiple efficacy scores are reported, resulting from replicated experiments. The response variable is a scalar value measuring knockdown efficacy from a one-hot encoded target sequence. Note that here we report results only on the de-noised averaged data across replicates (see Stirn et al. [2023]), while we leave the results on the raw replicates for App. D. Results on these datasets validate the advantage of using the posterior predictive for our methods, as well as the effectiveness of using the natural parameterization, which results in improved training stability.

## 5.3 Regression with Image Data

To assess the performance of different approaches in a typical deep learning setting, we create versions of the MNIST and FashionMNIST datasets where the task is to regress the rotational angle under label-dependent heteroscedastic noise (Problem 2.1). Such a setting is interesting to study since most (heteroscedastic) regression datasets are tabular and we intend to benchmark on a more common deep learning task where tabular regression models would not be applicable.

| Objective | Regularization | Posterior Predictive | MNIST with MLP | | | FashionMNIST with CNN | | |
|---|---|---|---|---|---|---|---|---|
| | | | LL ($\uparrow$) | $D_{\mathrm{KL}}$ ($\downarrow$) | RMSE ($\downarrow$) | LL ($\uparrow$) | $D_{\mathrm{KL}}$ ($\downarrow$) | RMSE ($\downarrow$) |
| Homoscedastic | EB | ✓ | -5.55 (0.01) | 0.77 (0.02) | 25.6 (0.8) | -5.54 (0.00) | 0.67 (0.00) | 18.7 (0.2) |
| Naive NLL | GS | ✗ | -5.56 (0.01) | 0.68 (0.00) | 19.0 (0.1) | -5.37 (0.01) | 0.49 (0.00) | 20.1 (0.1) |
| $\beta$-NLL (0.5) | GS | ✗ | -5.36 (0.01) | 0.49 (0.01) | 20.7 (0.8) | -5.47 (0.03) | 0.59 (0.03) | 23.5 (0.3) |
| $\beta$-NLL (1) | GS | ✗ | -5.38 (0.01) | 0.51 (0.01) | 21.5 (0.5) | -5.53 (0.03) | 0.65 (0.03) | 25.5 (0.2) |
| Faithful | GS | ✗ | -5.56 (0.01) | 0.69 (0.00) | 21.1 (0.3) | -5.78 (0.00) | 0.91 (0.00) | 51.8 (0.0) |
| MC-Dropout | GS | ✓ | -5.42 (0.02) | 0.55 (0.01) | 21.0 (0.5) | -5.37 (0.01) | 0.48 (0.00) | 20.1 (0.1) |
| VI | GS | ✓ | -5.42 (0.04) | 0.56 (0.04) | 21.9 (0.4) | -5.39 (0.01) | 0.51 (0.01) | 21.4 (0.5) |
| Naive NLL | EB | ✓ | **-5.30 (0.00)** | **0.44 (0.01)** | **18.2 (0.3)** | **-5.34 (0.00)** | **0.47 (0.01)** | **18.3 (0.2)** |
| | | ✗ | -5.31 (0.01) | | | **-5.35 (0.01)** | | |
| Natural NLL | GS | ✓ | -5.42 (0.00) | 0.55 (0.00) | 19.2 (0.1) | -5.40 (0.01) | 0.56 (0.01) | 24.3 (1.4) |
| | | ✗ | -5.42 (0.00) | | | -5.44 (0.01) | | |
| | EB | ✓ | **-5.30 (0.01)** | **0.45 (0.01)** | 19.1 (0.2) | **-5.34 (0.00)** | **0.46 (0.00)** | **18.1 (0.1)** |
| | | ✗ | -5.32 (0.01) | | | **-5.35 (0.00)** | | |

Table 2: Performance metrics on heteroscedastic image regression (Problem 2.1) on MNIST and FashionMNIST with MLP and CNN architectures, respectively. We report the mean and standard error and bold all numbers that are statistically indistinguishable from the best result. The bottom half shows the proposed methods. We find that the empirical Bayes (EB) procedure leads to better regularized and performing models than using grid search (GS). The posterior predictive also strictly improves the test log likelihood, albeit not as significantly.

To create the modified datasets, we follow the procedure as outlined in Problem 2.1. An observation $\mathbf{x}$ is generated by rotating an MNIST (resp. FashionMNIST) image with rotation angle drawn as $\mathtt{rot} \sim \mathrm{Unif}(-90, 90)$, and the corresponding target $y$ is generated as $y = \mathtt{rot} + (11c + 1)\epsilon$, where $c \in \{0, 1, \ldots, 9\}$ is the image class of $\mathbf{x}$ as integer number and $\epsilon \sim \mathcal{N}(0, 1)$ is an independent noise source. To solve the problem, a heteroscedastic regression model needs to 1) learn the mapping of rotated image to angle and 2) learn the label of the image to correctly fit the heteroscedasticity of the observation noise, both of which require learning complex functions. We evaluate the methods based on the test log likelihood and RMSE. In addition, we compute the KL-divergence of the predicted distribution from the ground truth.[4] We train a 3-layer 500-neuron wide MLP with ReLU activation for the MNIST-based dataset and a CNN (3 conv. and 2 linear) for the task based on FashionMNIST. In App. D.2.4, we provide additional details and results on two alternative tasks.

The results on image regression with heteroscedastic label noise indicate that the proposed empirical Bayes (EB) algorithm leads to better generalizing models and the posterior predictive (PP) strictly improves performance for both parameterizations. It is also notable that *Faithful* fails to fit the model due to the restrictions on its variance function, and therefore performs worse than a homoscedastic model on FashionMNIST. $\beta$-NLL, MC-Dropout, and VI all give relatively good results but fall slightly short of the proposed methods.

## 6 Conclusions

We tackled the problem of heteroscedastic regression using neural networks with a focus on regularization. To this end we proposed three individual improvements: 1) we use the natural form of the Gaussian likelihood; we derive a scalable Laplace approximation that 2) can be used for automatic principled regularization without validation data, and 3) enables epistemic uncertainty estimation through its posterior predictive. We show on image data that our method is scalable despite the Bayesian procedures and benchmark our approach on UCI regression as well as CRISPR-Cas13 datasets achieving state-of-the-art performance. For future work, it would be interesting to relax the assumption about Gaussianity and apply our approach to other real-world datasets.

---

[4]Note, however, that the original images are not completely free of rotation which creates an additional, potentially heteroscedastic, noise source that we cannot quantify.

**Acknowledgements**

AI gratefully acknowledges funding by the Max Planck ETH Center for Learning Systems (CLS). EP and AM were supported by a fellowship from the ETH AI Center.

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

# Appendix

## Table of Contents

# A  Laplace with Mean and Variance Parametrization

In Sec. 4, we introduced the Laplace approximation for the naturally parameterized Gaussian likelihood. The natural parametrization allows an efficient generalized Gauss-Newton approximation to the Hessian due to the properties of the second derivative [Martens, 2020], i.e., that is positive definite w.r.t. the natural parameters. The positive definiteness is necessary so that the generalized Gauss-Newton, which pre and post-multiplies the Jacobians to this Hessian, can only be positive semidefinite. This is a necessary requirement for the Laplace approximation.

The Laplace approximation provides different useful capabilities for heteroscedastic regression: 1) automatic regularization through marginal likelihood optimization (empirical Bayes), 2) an approximation to the posterior predictive, and 3) epistemic uncertainties on the functional output. While we derived these in the context of the natural parametrization in the main text, it is also possible to obtain these benefits for the common mean-variance parametrization of a neural network.

Since the Hessian of the Gaussian likelihood under the mean-variance parametrization is not positive definite, we cannot directly use this parametrization for the Gauss-Newton approximation. To verify this statement, define $\mathbf{m}(x;\boldsymbol{\theta}) = \left[\mu(\mathbf{x};\boldsymbol{\theta}), \sigma^2(\mathbf{x};\boldsymbol{\theta})\right]^\top$ and compute the Hessian of the Gaussian negative log likelihood w.r.t. $\mathbf{m}$

$$-\nabla_{\mathbf{m}}^2 \log p(y|\mathbf{x},\boldsymbol{\theta}) = \begin{bmatrix} \frac{1}{\sigma^2(\mathbf{x};\boldsymbol{\theta})} & \frac{y-\mu(\mathbf{x};\boldsymbol{\theta})}{(\sigma^2(\mathbf{x};\boldsymbol{\theta}))^2} \\ \frac{y-\mu(\mathbf{x};\boldsymbol{\theta})}{(\sigma^2(\mathbf{x};\boldsymbol{\theta}))^2} & \frac{(\mu(\mathbf{x};\boldsymbol{\theta})-y)^2 - \frac{\sigma^2(\mathbf{x};\boldsymbol{\theta})}{2}}{(\sigma^2(\mathbf{x};\boldsymbol{\theta}))^3} \end{bmatrix} \tag{14}$$

By Silvester's criterion, a Hermitian matrix is positive definite if and only if the determinants of all sub-matrices are positive. For the Hessian in Equation 14, this criterion is not fulfilled as

$$\det\left(-\nabla_{\mathbf{f}}^2 \log p(y|\mathbf{x},\boldsymbol{\theta})\right) = -\frac{1}{2(\sigma^2(\mathbf{x};\boldsymbol{\theta}))^3} < 0.$$

A simple way to overcome this issue is to map the mean-variance parameters $\mathbf{m}$ to the natural parameters $\boldsymbol{\eta}$ using Equation 3 and apply the approximations as derived in Sec. 4, i.e.,

$$\eta_1(\mathbf{x};\boldsymbol{\theta}) = \frac{\mu(\mathbf{x};\boldsymbol{\theta})}{\sigma^2(\mathbf{x};\boldsymbol{\theta})} \text{ and } \eta_2(\mathbf{x};\boldsymbol{\theta}) = -\frac{1}{2\sigma^2(\mathbf{x};\boldsymbol{\theta})}, \tag{15}$$

and then apply the Gauss-Newton approximation as in Sec. 4 to obtain

$$-\nabla_{\mathbf{m}}^2 \log p(y|\mathbf{x},\boldsymbol{\theta}) \approx -J_{\boldsymbol{\eta}|\mathbf{m}}^\top \nabla_{\boldsymbol{\eta}}^2 \log p(y|\mathbf{x},\boldsymbol{\theta}) J_{\boldsymbol{\eta}|\mathbf{m}} = \begin{bmatrix} \frac{1}{\sigma^2(\mathbf{x};\boldsymbol{\theta})} & 0 \\ 0 & \frac{1}{2(\sigma^2(\mathbf{x};\boldsymbol{\theta}))^2} \end{bmatrix}, \tag{16}$$

where the Jacobians $J_{\boldsymbol{\eta}|\mathbf{m}}$ are of $\boldsymbol{\eta}$ w.r.t. $\mathbf{m}$. It is easy to see that the Hessian approximation in Equation 16 is positive definite by evaluating the determinants of all sub-matrices.[5] The implementation is very simple: we simply use the output of the neural network $\mathbf{m}$, apply the transformation to the natural parameters $\boldsymbol{\eta}$, and then compute the quantities as derived for the natural Laplace approximation.

# B  Comparison of Gradient Updates

In the following, we first review the problem of the mean-variance parametrization identified by Seitzer et al. [2022] and present their proposed solution. Then we recap the proposal by Stirn et al. [2023], and subsequently explain how the natural parametrization avoids the gradient scaling problem.

**The Problem**  Seitzer et al. [2022] identify that a problem of previous neural estimators for heteroscedastic regression is due to the gradient of the negative log likelihood (NLL) which are parametrized to estimate the mean and standard deviation. First, recap that the corresponding NLL with respect to neural network parameters $\boldsymbol{\theta}$ can be written as

$$\ell_{\mu,\sigma}(\boldsymbol{\theta}) = \tfrac{1}{2}\log\sigma^2(\mathbf{x};\boldsymbol{\theta}) + \frac{(y-\mu(\mathbf{x};\boldsymbol{\theta}))^2}{2\sigma^2(\mathbf{x};\boldsymbol{\theta})} + const. \tag{17}$$

---

[5]Interestingly, the individual determinants for the $1 \times 1$ and $2 \times 2$ sub-matrices are equal to the absolute value of the corresponding determinants of $-\nabla_{\mathbf{f}}^2 \log p(y|\mathbf{x},\boldsymbol{\theta})$ reminiscent of saddle-free methods.

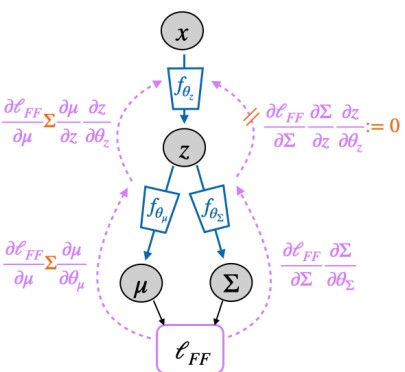

Figure 4: Illustration for the faithful heteroscedastic regression method proposed by Stirn et al. [2023]. Functions parameterized by neural networks are color-coded in blue, backpropagated gradients are color-coded in violet, and the two black arrows only indicate that the values for $\mu$ and $\Sigma$ are used as input for loss computation. The two modifications to conventional heteroscedastic regression introduced by Stirn et al. [2023] are color-coded in orange: the introduction of the scaling of the gradient $\nabla_\mu \ell_{\text{FF}}(\boldsymbol{\theta})$ by the covariance, and of a stop gradient operation to avoid $\nabla_\Sigma \ell_{\text{FF}}(\boldsymbol{\theta})$ gradients being propagated to $f_{\theta_{\mathbf{z}}}$. In this setup, specific dependencies for $\Sigma$ on $\mathbf{x}$ which are independent of the mean function are discarded, especially in the case in which $\mathbf{z}$ acts as a bottleneck. Note that in this image we make the dependency of neural network functions on their respective parameters explicit. In the rest of the work we omit the $\theta$ for the sake of brevity and readability.

As Seitzer et al. [2022] point out, the gradient of $\ell_{\mu,\sigma}(\boldsymbol{\theta})$ with respect to $\mu(\mathbf{x}; \boldsymbol{\theta})$ is scaled by the estimated precision, i.e,

$$\nabla_\mu \ell_{\mu,\sigma}(\boldsymbol{\theta}) = \frac{\mu(\mathbf{x};\boldsymbol{\theta})-y}{\sigma^2(\mathbf{x};\boldsymbol{\theta})}, \quad \nabla_{\sigma^2} \ell_{\mu,\sigma}(\boldsymbol{\theta}) = \frac{\sigma^2(\mathbf{x};\boldsymbol{\theta})-(y-\mu(\mathbf{x};\boldsymbol{\theta}))^2}{2(\sigma^2(\mathbf{x};\boldsymbol{\theta}))^2} \tag{18}$$

which they argue can lead to a poor mean estimate. In particular, they state that if the variance is well-calibrated, i.e., $\sigma^2(\mathbf{x}; \boldsymbol{\theta}) \approx (\mu(\mathbf{x}; \boldsymbol{\theta}) - y)^2$, the gradient with respect to the mean reduces to

$$\nabla_\mu \ell_{\mu,\sigma}(\boldsymbol{\theta}) \approx \frac{\mu(\mathbf{x};\boldsymbol{\theta})-y}{\sigma^2(\mathbf{x};\boldsymbol{\theta})} = \frac{1}{\mu(\mathbf{x};\boldsymbol{\theta})-y} \tag{19}$$

which rewards points with an already accurate mean fit and undersamples points for which the mean fit is inaccurate [Seitzer et al., 2022]. We would like to note, however, that the same holds if the irreducible aleatoric uncertainty is low resp. high in a certain region, which does not necessarily imply a poor mean fit.

$\beta$-**NLL**    To improve the mean fit, Seitzer et al. [2022] introduce the $\beta$-NLL loss

$$\ell_\beta(\boldsymbol{\theta}) = \lfloor \sigma^{2\beta}(\mathbf{x};\boldsymbol{\theta}) \rfloor \left( \tfrac{1}{2} \log \sigma^2(\mathbf{x};\boldsymbol{\theta}) + \frac{(y-\mu(\mathbf{x};\boldsymbol{\theta}))^2}{2\sigma^2(\mathbf{x};\boldsymbol{\theta})} + c \right) \tag{20}$$

which introduces a stop-gradient operation denoted as $\lfloor \cdot \rfloor$ to trade-off the influence of the variance estimate on the gradients. The corresponding gradients for the $\beta$-NLL loss are

$$\nabla_\mu \ell_\beta(\boldsymbol{\theta}) = \frac{\mu(\mathbf{x};\boldsymbol{\theta})-y}{\sigma^{2-2\beta}(\mathbf{x};\boldsymbol{\theta})}, \quad \nabla_{\sigma^2} \ell_\beta(\boldsymbol{\theta}) = \frac{\sigma^2(\mathbf{x};\boldsymbol{\theta})-(y-\mu(\mathbf{x};\boldsymbol{\theta}))^2}{2\sigma^{4-2\beta}(\mathbf{x};\boldsymbol{\theta})} . \tag{21}$$

Thus, for $\beta = 1$, the gradient for the mean is proportional to the gradient for the homoscedastic objective (which does not hold for the gradient of the variance), for $\beta = 0$ it is the standard heteroscedastic gradient and for $0 < \beta < 1$ the approach interpolates between both extremes. A special case appears at $\beta = 0.5$, where the gradient of the mean is weighted by the standard deviation.

**Proposal of [Stirn et al., 2023]**    To approach the gradient scaling, Stirn et al. [2023] argue that a heteroscedastic regression model that learns $\mu(\mathbf{x}), \sigma(\mathbf{x})$ should be faithful to a homoscedastic model that learns $\mu_0(\mathbf{x})$. As faithfulness they define that (cf. Stirn et al. [2023, Def. 2])

$$\mathbb{E}_{(\mathbf{x},y)\sim\mathcal{D}}[(y - \mu_0(\mathbf{x}))^2] \geq \mathbb{E}_{(\mathbf{x},y)\sim\mathcal{D}}[(y - \mu(\mathbf{x}))^2] . \tag{22}$$

To achieve this property, they propose to decouple the estimation into three networks, as shown in Figure 4: a shared representation learner $f_{\mathbf{z}}$ computes a representation $\mathbf{z}$ from $\mathbf{x}$, which is passes into two individual networks $f_\mu$ and $f_\Sigma$, which receive $\mathbf{z}$ as input and output the mean respectively the covariance matrix. Based on this network architecture, they minimize the following loss

$$\ell_{\mathrm{FF}}(\boldsymbol{\theta}) = \tfrac{(y-\mu(\mathbf{x};\boldsymbol{\theta}))^2}{2} - \log \mathcal{N}\left(y \,|\, \lfloor (f_\mu \circ f_{\mathbf{z}})(\mathbf{x}) \rfloor, \Sigma(\lfloor f_{\theta_z}(\mathbf{x}) \rfloor)\right) , \tag{23}$$

which incorporates two stop-gradient operations. They show that the proposal is equivalent to the following two modifications, which are illustrated in Figure 4: First, they stop the gradient of $f_\Sigma$ from propergating to $f_{\mathbf{z}}$. Second, they propose to multiply the gradient of $f_\mu \circ f_{\mathbf{z}}$ with $\Sigma$. In combination, both modifications imply that the gradient with respect to the mean is equivalent to the homoscedastic model [Stirn et al., 2023]. As discussed in Sec. 2 and App. C, these modifications restrict the capacity of the joint network such that it cannot pick-up complex dependencies for the variance that do not affect the mean.

Last, the procedure has some commonalities to $\beta$-NLL. In particular, for the sub-networks $f_{\mathbf{z}}$ and $f_\mu$ the gradients are multiplied with $\Sigma$ which is the multivariate analogue of the effect of the $\beta$-NLL loss on the mean gradient for $\beta = 1$.

**Proposed: Natural Parametrization**   The natural parametrization, that we use in this paper, has the following gradients with respect to $\ell_{\boldsymbol{\eta}}(\boldsymbol{\theta})$ with respect to $\eta_1$ and $\eta_2$ as

$$\nabla_{\eta_1}\ell_{\boldsymbol{\eta}}(\boldsymbol{\theta}) = -\tfrac{\eta_1(\mathbf{x};\boldsymbol{\theta})}{2\eta_2(\mathbf{x};\boldsymbol{\theta})} - y, \quad \nabla_{\eta_2}\ell_{\boldsymbol{\eta}}(\boldsymbol{\theta}) = \tfrac{(\eta_1(\mathbf{x};\boldsymbol{\theta}))^2}{4(\eta_2(\mathbf{x};\boldsymbol{\theta}))^2} - \tfrac{1}{2\eta_2(\mathbf{x};\boldsymbol{\theta})} - y^2 \tag{24}$$

Although these gradients do not have a direct implication on updates with respect to mean and variance, note that if we compute the mean estimate $\mu(\mathbf{x};\boldsymbol{\theta})$ from the estimated natural parameters, it is equal to $-\tfrac{\eta_1(\mathbf{x};\boldsymbol{\theta})}{2\eta_2(\mathbf{x};\boldsymbol{\theta})}$. Similarly, note that the variance $\sigma^2(\mathbf{x};\boldsymbol{\theta})$ can be computed from the natural parametization as $-\tfrac{1}{2\eta_2(\mathbf{x};\boldsymbol{\theta})}$. If we replace the corresponding quantities in Equation 24, we get that

$$\nabla_{\eta_1}\ell_{\boldsymbol{\eta}}(\boldsymbol{\theta}) = \mu(\mathbf{x};\boldsymbol{\theta}) - y, \quad \nabla_{\eta_2}\ell_{\boldsymbol{\eta}}(\boldsymbol{\theta}) = \sigma^2(\mathbf{x};\boldsymbol{\theta}) - (y^2 - (\mu(\mathbf{x};\boldsymbol{\theta}))^2) , \tag{25}$$

which is desired according to Seitzer et al. [2022] and Stirn et al. [2023] since these are simply separate residuals for mean and variance. Empirically, we observe that this parametrization can be more stable to train and less prone to insufficient regularization, which is in line with previous work on Gaussian processes and ridge regression [Le et al., 2005, Immer et al., 2023a].

## C   Orthogonal Mean-Variance Sources

To illustrate the limitations of using a joint encoder network for mean and variance, which receives no gradients from the variance estimate (cf. [Stirn et al., 2023, *Faithful*]) we consider a toy setting in which mean and variance are generated from independent random variables:

**Problem C.1**   *Let $\mathbf{X} \in \mathbb{R}^2$ be a centered multivariate Gaussian distributed with $\Sigma = I$. We construct the target variable $Y$ as*

$$y = \sin(x_1) + \tfrac{1}{2}\left(|x_2| + 0.2\right)\epsilon$$

*where $\epsilon \sim \mathcal{N}(0,1)$ is an independent noise variable.*

From the mechanism described in Problem C.1, we construct 20 datasets consisting of 1000 i.i.d. samples, where each dataset is generated using a different random seed. We split the data into train-validation/test according to the ratio (0.9/0.1). Then the train-validation data are split into training and validation sets (0.9/0.1).

As baselines we consider our empirical Bayes training with the natural homoscedastic and heteroscedastic objective using only a point predictive, both trained with the full Laplace approximation, where we use an MLP with 1 hidden-layer consisting of 100 hidden units. For *Faithful*, we use the same MLP with 1 hidden layer for the joint encoder $f_{\mathbf{z}}$ and instantiate $f_\mu$ and $f_\Sigma$ as linear projections from dimension $|\mathbf{z}|$ to 1. We ablate the size of the hidden dimension $|\mathbf{z}|$ from 1 to 128 neurons to create a bottleneck. In theory, a size of 2 should be sufficient to encode both mean and variance.

As shown in Figure 5, the bottleneck size has an influence on how much information about the variance is propagated, however, a large bottleneck size does not guarantee that the network learns

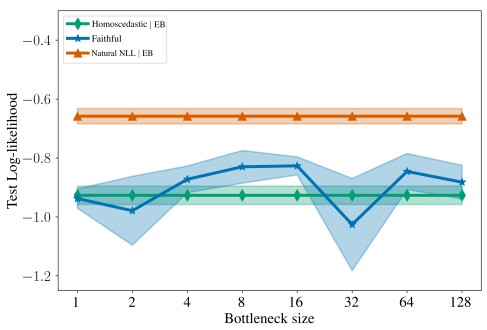
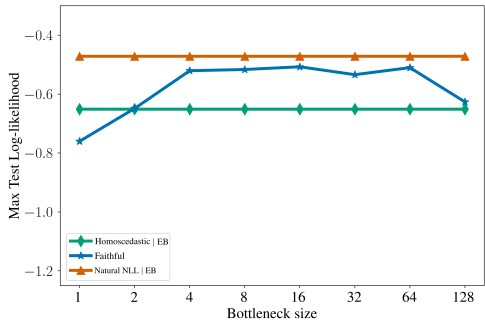

(a) Average test log likelihood across seeds      (b) Best test log likelihood across seeds

Figure 5: Visualization of the results for data generated according to Problem C.1, where only *Faithful* is ablated with respect to the bottleneck size. Left: Average test log likelihood for each of the compared models, with standard error as error bars. Right: Best test log likelihood result achieved across seeds for each of the models.

the relevant information about the variance. We rather conjecture that the amount of information that is propagated is dependent on the initialization of the network, since no gradient actively encourages the network $f_{\mathbf{z}}$ to pick up information about $X_2$, which influences the variance. Since for a larger bottleneck size the network can encode much more information than just $X_1$, we observe that it encodes information about $X_2$ more consistently, and for some seeds (see Figure 5b) it performs comparably well with our heteroscedastic variant. On average, however, its performance lies, as expected, between the homoscedastic and heteroscedastic models with natural parametrization trained with the full Laplace approximation.

# D    Details on Experiments and Additional Results

In this section, we first provide technical details for our experimental comparisons, and then show some additional results for our experiments.

## D.1    Experimental Details

### D.1.1    Skafte Illustrative Example

In Figure 1, we compare a homoscedastic to a heteroscedastic likelihood and the point-wise prediction to the Laplace-approximated posterior predictive. The dataset is created similar to Skafte et al. [2019] with the following generative model:

$$y = \sin(x) + (0.1 + |0.5x|)\varepsilon \quad \text{with} \quad x \sim \text{Unif}[2.5, 12.5] \quad \text{and} \quad \varepsilon \sim \mathcal{N}(0, 1). \quad (26)$$

For training, we center the input data by subtracting the mean $7.5$ from $x$ and standardize the observations to zero mean and unit variance as common for regression problems. We create a dataset with 1000 samples.

We model the data with a neural network with 100 hidden units on a single hidden layer using `tanh` activation. Both homoscedastic and heteroscedastic models are trained using the marginal-likelihood optimization algorithm [Immer et al., 2021a], also detailed in Alg. 1. We train the neural network parameter using learning rate $0.01$ with Adam [Kingma and Ba, 2014] and optimize the hyperparameters using learning rate $0.1$ with the same optimizer every 50 epochs for 50 gradient steps. In this example, we use the efficient Kronecker-factored Laplace approximation and use the model that obtained the best marginal likelihood approximation during training, akin to early stopping.

### D.1.2    UCI Regression and CRISPR-Cas13 Knockdown Efficacy Experiments

For both UCI and CRISPR regression datasets all compared models use an MLP with 1 hidden layer, with width of 50 units. We use the `GeLU` activation function for models trained on the UCI regression datasets, while we adopt the `ReLU` activation function for models trained on the CRISPR datasets. We train all models, except for the VI and MC-Dropout baselines, with Adam optimizer using a

batch size of 256 for 5000 epochs and an initial learning rate of $10^{-2}$ that is decayed to $10^{-5}$ using a cosine schedule. As we observe that gives overall better performance, MC-Dropout and VI on UCI datasets are trained for 1000 epochs, with a learning rate of $10^{-3}$ that is not decayed. In the models using classic grid-search (GS) for hyperparameter tuning the grid-search procedure is used to tune the prior precision, i.e., weight decay, hyperparameter. On the CRISPR datasets, VI is trained for 500 epochs with a learning rate of $10^{-3}$. For all methods using grid-search, we first split the training data into a 90/10 train-validation split. We then train the model for each order of magnitude of the prior precision from $10^{-3}$ to $10^5$. The best prior precision is selected on the validation data and the model is retrained on the entire combined training dataset and assessed on the test data. The observation noise for the homoscedastic model is chosen on the training data as maximum likelihood solution. With the Empirical Bayes (EB) training approach, we only need to fit the model once on the entire training data and optimize the prior precision values per layer of the neural network during training. We use the algorithm in Immer et al. [2021a] with a hyperparameter learning rate of 0.01 decayed to 0.001 using a cosine schedule, 100 burn-in epochs, every 50 epochs for 50 hyperparameter steps. The pseudocode for the optimization is provided in App. E.1. For the UCI regression datasets, we use the full Hessian Laplace approximation, while for CRISPR dataset, due to higher computational cost of training, we resort to the Kronecker-factored Laplace approximation. Finally for MC-Dropout, we set the dropout probability for dropout layers to 0.005. For all compared models, results are averaged over 20 independent runs on the UCI regression datasets and 10 independent runs on the CRISPR datasets, where independent runs use different seeds. We report standard errors. Runs were executed on a computing cluster, but without the need for GPU support. Note that in our experiments to implement the VI baseline we make use of the PyTorch implementation from Krishnan et al. [2022].

### D.1.3 Image Regression

We use the same hyperparameters for MNIST and FashionMNIST but two different architectures. The architecture for MNIST is an MLP with 3 hidden layers, each with width of 500 units and using `ReLU` activation function. The architecture for FashionMNIST is a LeNet with three convolutional layers with $6, 32, 256$ channels, respectively, with max-pooling in-between followed by two hidden layers of width 128 and 64. In all cases except for VI and MC-Dropout, we train the models with SGD using a batch size of 512 for 500 epochs and initial learning rate of 0.1 that is decayed to $10^{-4}$ using a cosine schedule. For MC-Dropout, we use a learning rate of 0.01 and train the model for 50 epochs with the Adam optimizer, with dropout probability for Dropout layers of 0.005. For VI, we use a learning rate of $10^{-2}$ that is decayed to $10^{-4}$ using a cosine schedule when training on the FashionMNIST dataset, while we use a non-decayed learning rate of $10^{-2}$ for the MNIST dataset. In both cases, the VI baseline is trained for 500 epochs with the Adam optimizer. In practice, we observe that the VI and MC-Dropout baselines are more sensitive to choices of hyperparameters than the other compared models, and we adopted the settings that would lead to the best results according to our experiments. In the models using classic grid-search (GS) for hyperparameter tuning (GS) the grid-search procedure is used to tune the prior precision, i.e., weight decay, hyperparameter. For all methods using grid-search, we first split the training data into a 90/10 train-validation split. We then train the model for each order of magnitude of the prior precision from $10^{-2}$ to $10^4$. The best prior precision is selected on the validation data and the model is retrained on the entire combined training dataset and assessed on the test data. The observation noise for the homoscedastic model is chosen on the training data as maximum likelihood solution. With the Empirical Bayes (EB) training approach, we only need to fit the model once on the entire training data and optimize the prior precision values per layer of the neural network during training. We use the algorithm in Immer et al. [2021a] with a hyperparameter learning rate of 0.1, 50 burn-in epochs, every 25 epochs for 50 hyperparameter steps. These choices are in line with their suggestions. Due to the large models on the image datasets, we use the Kronecker-factored Laplace approximation so that the training and inference time between Laplace and baselines are almost identical. The training was done 5 times (different seeds) per model-dataset pair to estimate mean and standard error and were run on a computing cluster with V100 and A100 NVIDIA GPUs.

## D.2    Additional Experimental Results

### D.2.1    Skafte Illustrative Example

In Figure 7, we additionally compare the proposed approach to Bayesian neural network baselines, MC-dropout and mean-field VI. We find that the proposed heteroscedastic Laplace approximation with natural parameterization provides more visually appealing in-between epistemic uncertainties and has a more sensible split into aleatoric and epistemic uncertainties. In Figure 6, we display the difference when the generated data were homoscedastic: despite this, our well-regularized heteroscedastic method performs on par or better with a homoscedastic baseline trained in an identical way.

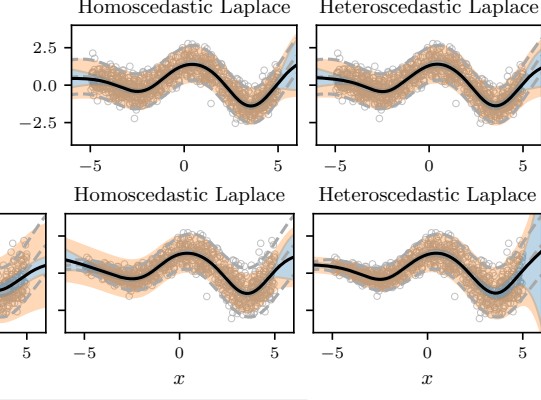

Figure 7: Comparison of heteroscedastic Laplace (right) to Monte-Carlo Dropout [Gal and Ghahramani, 2016], mean-field variational inference, and homoscedastic Laplace on the Skafte [Skafte et al., 2019] regression example. Laplace obtains better "in-between" uncertainties as reported also by Foong et al. [2019].

### D.2.2    UCI Regression Datasets

In this section, we report additional results to the ones we show in Sec. 5.1 for UCI regression datasets. In Table 3 we report test Mean Squared Error (MSE) results for the models compared in Table 1, while Table 4 reports an ablation where for our proposed natural parameterized NLL heteroscedastic loss model EB regularization, comparing results obtained with the full Hessian Laplace approximation (*full*) and results obtained with the Kronecker-factored Laplace approximation (*kfac*). Results show that the full approximation can improve results, and thus should be preferred when feasible. At the same time, the *kfac* approximation leads to comparable results in most settings, justifying its use for larger datasets. In Table Table 5, we compare the results obtained with two different variants of posterior predictive for the Natural NLL (EB) loss on the UCI regression datasets, namely the closed-form posterior predictive proposed in Sec. 4.4 (*mean*) and the asymptotically exact Monte-Carlo estimate of the NLL [Tomczak et al., 2018] (*lse*). Finally Table 6 shows that the results from our work for models trained with EB regularization favourably compare to results from the work of Wu et al. [2019] on a deterministic variational inference approach for BNNs that also adopts an EB procedure for regularization. Note however, that this last approach does not scale well to more complex architectures than an MLP and hence is not reported in the other experimental settings.

| Objective | Regular-ization | Posterior Predictive | LL (↑) | | | | | | | |
|---|---|---|---|---|---|---|---|---|---|---|
| | | | *boston* | *concrete* | *energy* | *kin8nm* | *naval* | *plant* | *wine* | *yacht* |
| Homoscedastic | EB | ✓ | -2.51 (0.04) | -3.03 (0.03) | -0.71 (0.04) | 1.29 (0.01) | 5.58 (0.45) | -2.83 (0.02) | -0.94 (0.02) | -1.05 (0.14) |
| Deterministic VI | EB | ✗ | -2.41 (0.02) | -3.06 (0.01) | -1.01 (0.06) | 1.13 (0.00) | 6.29 (0.04) | -2.80 (0.00) | -0.90 (0.01) | -0.47 (0.03) |
| Naive NLL | EB | ✓ | -2.47 (0.04) | -2.88 (0.03) | -0.46 (0.03) | 1.37 (0.01) | 6.38 (0.16) | -2.79 (0.02) | -0.93 (0.02) | -0.05 (0.15) |
| | | ✗ | -6.03 (1.82) | -3.62 (0.26) | -0.97 (0.15) | 1.36 (0.01) | -10.9 (8.01) | -2.83 (0.04) | -2.26 (0.51) | -1.27 (0.51) |
| Naive NLL | EB | ✓ | -2.36 (0.03) | -2.93 (0.02) | -0.71 (0.03) | 1.36 (0.01) | 6.66 (0.01) | -2.76 (0.02) | -0.94 (0.02) | -0.51 (0.04) |
| | | ✗ | -2.51 (0.09) | -2.99 (0.04) | -0.65 (0.05) | 1.35 (0.01) | 6.68 (0.01) | -2.77 (0.02) | -1.00 (0.03) | -0.60 (0.08) |

Table 6: Comparison of our models trained with EB regularization, with the work of Wu et al. [2019], that also adopts an EB regularization procedure.

| | | MSE (↓) | | | | | | | |
|---|---|---|---|---|---|---|---|---|---|
| Objective | Regular-ization | *boston* | *concrete* | *energy* | *kin8nm* | *naval* | *plant* | *wine* | *yacht* |
| Homoscedastic | EB | 0.130 (0.012) | 0.096 (0.005) | 0.002 (0.000) | 0.064 (0.001) | 0.001 (0.000) | 0.057 (0.002) | 0.606 (0.024) | 0.002 (0.000) |
| Naive NLL | GS | 0.170 (0.020) | 0.130 (0.013) | 0.009 (0.005) | 0.072 (0.001) | 5.673 (5.216) | 0.056 (0.002) | 0.605 (0.022) | 0.003 (0.000) |
| $\beta$-NLL (0.5) | GS | 0.131 (0.013) | 0.122 (0.007) | 0.002 (0.000) | 0.069 (0.001) | 6.293 (5.367) | 0.055 (0.002) | 0.608 (0.023) | 0.002 (0.001) |
| $\beta$-NLL (1.0) | GS | 0.144 (0.012) | 0.108 (0.009) | 0.003 (0.001) | 0.069 (0.001) | 0.003 (0.000) | 0.057 (0.002) | 0.609 (0.022) | 0.002 (0.000) |
| Faithful | GS | 0.140 (0.015) | 0.097 (0.007) | 0.004 (0.001) | 0.070 (0.001) | 0.003 (0.001) | 0.055 (0.002) | 0.616 (0.023) | 0.002 (0.001) |
| MC-Dropout | GS | 0.131 (0.016) | 0.131 (0.006) | 0.057 (0.003) | 0.117 ( 0.004) | 0.145 (0.011) | 0.059 (0.002) | 0.606 (0.025) | 0.124 (0.038) |
| VI | GS | 0.131 (0.016) | 0.134 (0.006) | 0.056 (0.004) | 0.076 (0.002) | 0.146 (0.009) | 0.058 (0.002) | 0.645 (0.037) | 0.042 (0.012) |
| Naive NLL | EB | 0.136 (0.016) | 0.100 (0.006) | 0.002 (0.000) | 0.065 (0.001) | 0.027 (0.017) | 0.055 (0.002) | 0.621 (0.027) | 0.003 (0.001) |
| Natural NLL | GS | 0.111 (0.010) | 0.081 (0.008) | 0.002 (0.000) | 0.067 (0.001) | 0.001 (0.000) | 0.051 (0.002) | 0.612 (0.022) | 0.003 (0.001) |
| | EB | 0.102 (0.011) | 0.094 (0.007) | 0.002 (0.000) | 0.061 (0.001) | 0.001 (0.000) | 0.052 (0.002) | 0.652 (0.046) | 0.003 (0.000) |

Table 3: Test MSE results for the compared models on the UCI regression datasets, where we report mean and standard error (dataset names in italic).

| | | | | LL (↑) | | | | | | | |
|---|---|---|---|---|---|---|---|---|---|---|---|
| Objective | Regular-ization | Approximation | Posterior Predictive | *boston* | *concrete* | *energy* | *kin8nm* | *naval* | *plant* | *wine* | *yacht* |
| Natural NLL | EB | kfac | ✓ | -2.44 (0.04) | -3.05 (0.02) | -0.94 (0.02) | 1.36 (0.01) | 6.64 (0.01) | -2.76 (0.02) | -0.94 (0.02) | -0.69 (0.05) |
| | | full | ✓ | -2.36 (0.03) | -2.93 (0.02) | -0.71 (0.03) | 1.36 (0.01) | 6.66 (0.01) | -2.76 (0.02) | -0.94 (0.02) | -0.51 (0.04) |
| | | kfac | ✗ | -2.51 (0.07) | -3.04 (0.03) | -0.83 (0.03) | 1.35 (0.01) | 6.69 (0.01) | -2.76 (0.02) | -5.41 (3.54) | -0.64 (0.05) |
| | | full | ✗ | -2.51 (0.09) | -2.99 (0.04) | -0.65 (0.05) | 1.35 (0.01) | 6.68 (0.01) | -2.77 (0.02) | -1.00 (0.03) | -0.60 (0.08) |

Table 4: Ablation comparing test log likelihood results for the natural parametrization with empirical Bayes (*Natural Laplace*) adopting the full Hessian Laplace approximation (*full*) and the Kronecker-factored Laplace approximation (*kfac*).

| | | | LL (↑) | | | | | | | |
|---|---|---|---|---|---|---|---|---|---|---|
| **Method** | Regular-ization | Posterior Predictive Variant | *boston* | *concrete* | *energy* | *kin8nm* | *naval* | *plant* | *wine* | *yacht* |
| Natural NLL | GS | mean | -2.45 (0.04) | -2.92 (0.04) | -0.73 (0.05) | 1.32 (0.01) | 6.66 (0.01) | -2.76 (0.02) | -0.94 (0.02) | -0.29 (0.07) |
| | | lse | -2.46 (0.04) | -2.92 (0.04) | -0.74 (0.06) | 1.32 (0.01) | 6.66 (0.01) | -2.76 (0.02) | -0.94 (0.02) | -0.29 (0.07) |
| Natural NLL | EB | mean | -2.39 (0.04) | -2.92 (0.03) | -0.71 (0.03) | 1.36 (0.01) | 6.66 (0.01) | -2.76 (0.02) | -0.94 (0.02) | -0.51 (0.04) |
| | | lse | -2.36 (0.03) | -2.93 (0.02) | -0.71 (0.03) | 1.36 (0.01) | 6.66 (0.01) | -2.76 (0.02) | -0.94 (0.02) | -0.51 (0.04) |

Table 5: Alblation comparing the two variants of posterior predictive for Natural NLL objective with Empirical Bayes regularization: *mean* and *lse*.

### D.2.3 CRISPR Gene Knockdown Efficacy Datasets

Here we report additional results for the compared models in Sec. 5.2 on the CRISPR datasets for gene knockdown efficacy, reporting results for models trained on single experiment replicates. Therefore, we report test log likelihood performance for models trained on the available replicates for each of the three CRISPR datasets in Figure 8. Again, for each model in each dataset we run 10 independent seeds. As it is expected, these results are in line with, and complement, the ones shown in Figure 3.

### D.2.4 Image Regression

In Table 9, we additionally provide results with the exact same training setting as for Table 2 but with a homoscedastic generative process. In Table 8, we provide results for the same task but with heteroscedastic noise that depends on the mean output, i.e., the rotational angle. Overall, the pattern is the same: empirical Bayes and the posterior predictive strictly improve generalization performance and the natural parameterization seems to overall perform best and most consistently. In Table 7, we report the runtimes for training (EB vs GS) and testing (MAP vs. PP). Training with EB is much faster than using a grid search but the posterior predictive requires more runtime per sample.

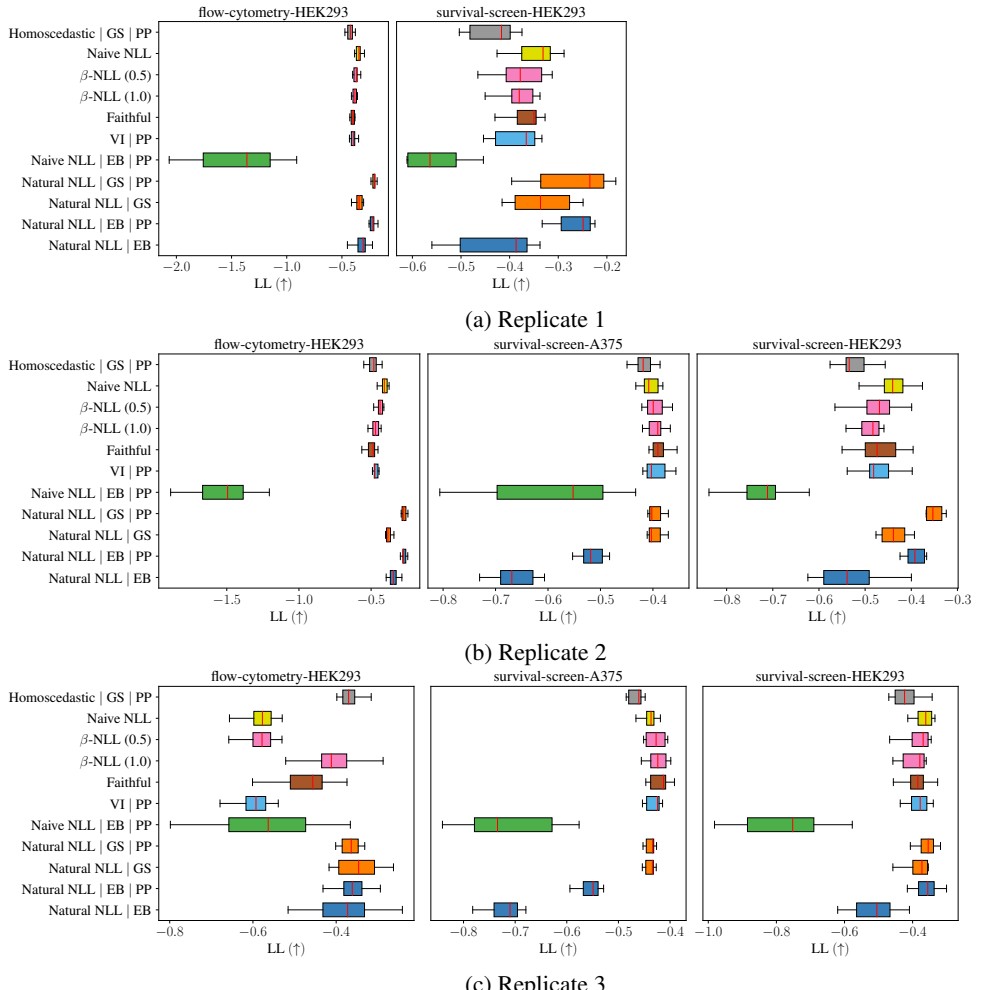

(a) Replicate 1

(b) Replicate 2

(c) Replicate 3

Figure 8: Box-plots reporting test log likelihood results on the CRISPR datasets on single experiment replicated. For the survival-screen-A375 dataset, only two replicates are available. Note that the NLL mean-variance parameterization with MAP prediction, as well as the MC-Dropout baseline, markedly underperform in this setting, and are hence not reported in the comparisons.

|  | $\beta$-NLL | Homoscedastic GS + PP | Homoscedastic EB + PP | Natural GS + PP | Natural EB + PP |
|---|---|---|---|---|---|
| Training | 44 min. | 44 min. | 8 min. | 49 min. | 9 min. |
| Inference | 0.023 sec. | 0.124 sec. | 0.124 sec. | 0.134 sec. | 0.134 sec. |

Table 7: Average runtime over 5 trials including hyperparameter optimization.

# E  Implementation

In the following, we detail the interleaved Laplace marginal likelihood optimization [Immer et al., 2021a] but adapted to the heteroscedastic setting in App. E.1. Further, we provide an efficient and stable implementation of the natural parametrization of the heteroscedastic Gaussian likelihood in App. E.2. We implement our method in the `laplace-torch` package [Daxberger et al., 2021] and the KFAC for heteroscedastic regression in the automatic second-order differentiation library [asdl; Osawa, 2021]. Due to anonymity reasons, we implemented these changes as private forks and will only propose them to the open source packages upon publication of the paper. This ensures that the proposed method can be used with any supported network by these packages and different scalable Laplace approximation variants.

| Objective | Regularization | Posterior Predictive | MNIST with MLP | | | FashionMNIST with CNN | | |
|---|---|---|---|---|---|---|---|---|
| | | | LL ($\uparrow$) | $D_{\mathrm{KL}}$ ($\downarrow$) | RMSE ($\downarrow$) | LL ($\uparrow$) | $D_{\mathrm{KL}}$ ($\downarrow$) | RMSE ($\downarrow$) |
| Homoscedastic | EB | ✓ | -4.02 (0.10) | 3.75 (1.92) | 11.9 (0.2) | -3.87 (0.02) | 1.37 (0.08) | **12.5 (0.2)** |
| Naive NLL | GS | ✗ | -4.56 (0.25) | 1.39 (0.25) | 20.9 (7.7) | -4.06 (0.16) | 0.88 (0.15) | 12.7 (0.2) |
| $\beta$-NLL (0.5) | GS | ✗ | -4.30 (0.16) | 1.13 (0.17) | 16.4 (1.9) | -4.01 (0.02) | 0.84 (0.02) | 15.0 (0.1) |
| $\beta$-NLL (1) | GS | ✗ | -4.21 (0.09) | 1.04 (0.09) | 15.0 (1.2) | -4.05 (0.04) | 0.88 (0.03) | **12.3 (0.1)** |
| Faithful | GS | ✗ | -4.17 (0.04) | 0.99 (0.04) | 14.1 (0.5) | -4.42 (0.05) | 1.25 (0.05) | 17.8 (1.6) |
| MC-Dropout | GS | ✓ | -4.14 (0.05) | 0.97 (0.05) | 15.9 (0.6) | -3.78 (0.05) | **0.61 (0.05)** | 15.5 (1.1) |
| VI | GS | ✓ | -4.15 (0.01) | 0.98 (0.01) | 13.9 (0.1) | -3.86 (0.03) | 0.69 (0.03) | 16.3 (0.7) |
| Naive NLL | EB | ✓ | -3.85 (0.02) | **0.75 (0.01)** | 12.2 (0.3) | **-3.66 (0.01)** | **0.60 (0.02)** | 13.1 (0.3) |
| | | ✗ | -3.92 (0.02) | **0.75 (0.01)** | 12.2 (0.3) | -3.77 (0.02) | **0.60 (0.02)** | 13.1 (0.3) |
| Natural NLL | GS | ✓ | -4.16 (0.01) | 0.99 (0.01) | 14.5 (0.1) | **-3.66 (0.01)** | **0.61 (0.02)** | 13.2 (0.4) |
| | | ✗ | -4.16 (0.01) | 0.99 (0.01) | 14.5 (0.1) | -3.78 (0.02) | **0.61 (0.02)** | 13.2 (0.4) |
| | EB | ✓ | **-3.73 (0.01)** | **0.76 (0.02)** | **11.0 (0.2)** | **-3.66 (0.01)** | **0.61 (0.01)** | 13.1 (0.2) |
| | | ✗ | -3.94 (0.02) | **0.76 (0.02)** | **11.0 (0.2)** | -3.78 (0.01) | **0.61 (0.01)** | 13.1 (0.2) |

Table 8: Image regression with heteroscedastic noise variance that depends on the rotation applied, i.e., $y \sim \mathrm{rot} + \sqrt{|\mathrm{rot}|}\varepsilon$. Similar to the results with heteroscedastic label-based noise in Table 2, empiricial Bayes and the posterior predictive improve the performance consistentlly.

| Objective | Regularization | Posterior Predictive | MNIST with MLP | | | FashionMNIST with CNN | | |
|---|---|---|---|---|---|---|---|---|
| | | | LL ($\uparrow$) | $D_{\mathrm{KL}}$ ($\downarrow$) | RMSE ($\downarrow$) | LL ($\uparrow$) | $D_{\mathrm{KL}}$ ($\downarrow$) | RMSE ($\downarrow$) |
| Homoscedastic | EB | ✓ | -4.15 (0.02) | 0.86 (0.07) | 12.8 (0.4) | -4.06 (0.01) | 0.64 (0.02) | **12.7 (0.2)** |
| Naive NLL | GS | ✗ | -5.38 (0.00) | 1.66 (0.00) | 51.8 (0.0) | -4.18 (0.03) | 0.45 (0.03) | **12.8 (0.2)** |
| $\beta$-NLL (0.5) | GS | ✗ | -4.34 (0.10) | 0.62 (0.10) | 16.0 (1.5) | -4.27 (0.02) | 0.54 (0.01) | 15.5 (0.1) |
| $\beta$-NLL (1) | GS | ✗ | -4.31 (0.05) | 0.59 (0.05) | 14.8 (0.8) | -4.26 (0.04) | 0.53 (0.03) | **12.6 (0.3)** |
| Faithful | GS | ✗ | -4.31 (0.03) | 0.58 (0.03) | 14.6 (0.5) | -4.57 (0.01) | 0.84 (0.01) | 21.0 (0.4) |
| MCDO | GS | ✓ | -4.29 (0.04) | 0.56 (0.04) | 16.1 (0.9) | -4.08 (0.02) | **0.36 (0.02)** | 15.6 (1.1) |
| VI | GS | ✓ | -4.30 (0.01) | 0.58 (0.01) | 14.5 (0.1) | -4.10 (0.01) | 0.38 (0.01) | 15.0 (0.3) |
| Naive NLL | EB | ✓ | -4.09 (0.01) | 0.42 (0.02) | 12.7 (0.1) | -4.02 (0.01) | **0.33 (0.01)** | 13.4 (0.2) |
| | | ✗ | -4.14 (0.02) | 0.42 (0.02) | 12.7 (0.1) | -4.06 (0.01) | **0.33 (0.01)** | 13.4 (0.2) |
| Natural NLL | GS | ✓ | -4.26 (0.00) | 0.54 (0.00) | 14.3 (0.1) | **-3.99 (0.01)** | **0.33 (0.01)** | **13.0 (0.2)** |
| | | ✗ | -4.26 (0.00) | 0.54 (0.00) | 14.3 (0.1) | -4.06 (0.01) | **0.33 (0.01)** | **13.0 (0.2)** |
| | EB | ✓ | **-4.06 (0.01)** | **0.38 (0.00)** | **12.0 (0.1)** | **-4.00 (0.01)** | **0.33 (0.01)** | 13.2 (0.2) |
| | | ✗ | -4.12 (0.01) | **0.38 (0.00)** | **12.0 (0.1)** | -4.06 (0.01) | **0.33 (0.01)** | 13.2 (0.2) |

Table 9: Image regression with homoscedastic noise variance that is fixed at scale of 10. Similar to the results with heteroscedastic label-based noise in Table 2, empiricial Bayes and the posterior predictive improve the performance consistently. Also, homoscedastic and $\beta$-NLL variants perform relatively well due to the reduced heteroscedasticity.

### E.1 Pseudo-Code for Our Method

Alg. 1 is the heteroscedastic regression variant of the algorithm proposed by Immer et al. [2021a] and includes the empirical Bayes hyperparameter optimization of the prior precision, or equivalently weight-decay, $\delta$ per layer. The algorithm optimizes neural network parameters $\theta$ and hyperparameters $\delta$ jointly during training and further the intermediate marginal likelihood values, $\log p(\mathcal{D}|\delta)$ can be used for early stopping. After training, either the Bayesian or point-wise posterior predictive can be used. The posterior predictive is efficient to compute and does not require sampling due to the simplification proposed in Sec. 4.4. Experimentally, we find that the Bayesian posterior predictive improves over the point-wise predictive in almost all cases.

---
**Algorithm 1** Optimization of Heteroscedastic Regression Models
---
**Require:** model $\mathbf{f}(\mathbf{x};\boldsymbol{\theta})$ (natural parameters), dataset $\mathcal{D}$, (initial) prior precision $\delta$, epochs $E$, burn-in $B$, frequency $F$, steps $S$, learning rate $\alpha$, hyperparameter learning rate $\gamma$

$\quad \forall l : \delta_l \leftarrow \delta$            $\triangleright$ Set prior precision (regularizer) for each layer

$\quad$ **for** epoch $e \leq E$ **do**

$\quad\quad$ **for** batch $\mathcal{B} \subseteq \mathcal{D}$ **do**        $\triangleright$ Optimize model parameters $\boldsymbol{\theta}$

$\quad\quad\quad \boldsymbol{\theta} \leftarrow \boldsymbol{\theta} + \alpha \nabla_{\boldsymbol{\theta}} \big[ \frac{|\mathcal{D}|}{|\mathcal{B}|} \log p(\mathcal{B}|\boldsymbol{\theta}) + \sum_l \log \mathcal{N}(\boldsymbol{\theta}_l|\mathbf{0}, \delta_l \mathbf{I}) \big]$    $\triangleright$ SGD parameter update

$\quad\quad$ **end for**

$\quad\quad$ **if** $e \mod F = 0$ and $e \geq B$ **then**       $\triangleright$ Optimize regularization $\boldsymbol{\delta}$

$\quad\quad\quad$ **for** step $s \leq S$ **do**

$\quad\quad\quad\quad \forall l : \delta_l \leftarrow \delta_l + \frac{\partial}{\partial \delta_l} \log p(\mathcal{D}|\boldsymbol{\delta})$ (Equation 12)    $\triangleright$ Hyperparameter update

$\quad\quad\quad$ **end for**

$\quad\quad$ **end if**

$\quad$ **end for**

$\quad$ **if** (Bayesian) posterior predictive **then**

$\quad\quad$ Compute $\mu(\mathbf{x}_*), \sigma^2(\mathbf{x}_*)$ as in Equation 13      $\triangleright$ Fast approximate posterior predictive

$\quad$ **else**

$\quad\quad \mu(\mathbf{x}_*) \leftarrow -\frac{\mathrm{f}_1(\mathbf{x}_*;\boldsymbol{\theta})}{2\mathrm{f}_2(\mathbf{x}_*;\boldsymbol{\theta})}$ and $\sigma^2(\mathbf{x}_*) \leftarrow -\frac{1}{2\mathrm{f}_2(\mathbf{x}_*;\boldsymbol{\theta})}$      $\triangleright$ Point-wise MAP predictive

$\quad$ **end if**

$\quad$ Predictive $\mathcal{N}(\mu(\mathbf{x}_*), \sigma^2(\mathbf{x}_*))$
---

## E.2   Implementation of Natural Gaussian Log Likelihood

Common automatic differentiation packages like pytorch [Paszke et al., 2017] and jax [Bradbury et al., 2018] do include typical losses and distributions but not the natural parametrization of a Gaussian. Below, we provide an implementation for it using pytorch:

```python
from math import log, pi
import torch

C = - 0.5 * log(2 * pi)

def heteroscedastic_mse_loss(input, target, reduction='mean'):
    """Heteroscedastic Normal negative log likelihood.

    Parameters
    ----------
    input : torch.Tensor (n, 2)
        two natural parameters per data point
    target : torch.Tensor (n, 1)
        targets
    """
    assert input.ndim == target.ndim == 2
    assert input.shape[0] == target.shape[0]
    n, _ = input.shape
    target = torch.cat([target, target.square()], dim=1)
    inner = torch.einsum('nk,nk->n', target, input)
    log_A = (input[:, 0].square() / (4 * input[:, 1])
             + 0.5 * torch.log(- 2 * input[:, 1]))
    log_lik = n * C + inner.sum() + log_A.sum()
    if reduction == 'mean':
        return - log_lik / n
    elif reduction == 'sum':
        return - log_lik
    else:
        raise ValueError('Invalid reduction', reduction)
```

