# OpenReview forum: "Effective Bayesian Heteroscedastic Regression with Deep Neural Networks"
_NeurIPS.cc/2023/Conference — NeurIPS 2023 poster_

### Official Review · Reviewer_7fHN · 2023-06-26

**Soundness:** 3 good
**Presentation:** 3 good
**Contribution:** 2 fair
**Rating:** 6
**Confidence:** 4

**Summary:**

The authors consider the task of properly modeling heteroscedastic observation noise with Bayesian Neural Networks (BNNs).
They identify several shortcomings in prior work concerning variance modeling and propose to rely on parameterization
via natural gradients instead. Together with a Laplace approximation to model epistemic uncertainty and a closed-form
posterior predictive likelihood approximation the proposed approach is evaluated on three sets of experiments.

**Strengths:**

- The paper considers a well-defined task, identifies prior weaknesses, and offers a principled solution.
- The paper is well written and can be understood easily by a reader
- The experimental setups cover a wide range of domains

**Weaknesses:**

- The individual steps, i.e., switching to natural parameters, reliance on Laplace approximations, and approximate
    closed-form marginalizations for the posterior predictive are mostly minor adaptations from prior work
- Section 5.2 reads more like an afterthought that is added to the paper to have an additional experiment.
    - The setup description is mostly left to the prior work which introduced the experiment.
    - The evaluation of the results is similarly vague. E.g., Heteroscedastic VI (PP) has a huge variance in Figure 1, Exp 1,
whereas it suffers no such problems in the individual replications. The Laplace approach of the proposed method
not only performs a little bit worse than the others in setting two (survival-screen-A375), but struggles a lot,
whereas the MAP approach does not suffer from this problem.


## Minor
- None of the tables do follow the NeurIPS style guide. Captions should be placed above a table, and tables should not contain vertical lines (fig 2)


## Typos
- Fig 2 right and Table 1/2 switch the se notation (from $\pm$ to brackets)
- Between Table 1 and 2 the formatting of the method column switches from right-aligned to left-aligned

**Questions:**

- I am not sure how to parse the sentence in l27/28. Hasn't epistemic uncertainty modeling been the main focus for now
    with aleatoric uncertainty modeling being of secondary interest? Can the authors provide some intuition on what they
    mean here? (I might just be missing the obvious...)
- Figure 2: How does the corresponding test NLL look like for the proposed approach on this task?
- On the VI setups:
    - Can the authors speculate on the performance of VI in the naval and plant experiments? In both data sets the
            performance is not just a drastic outlier compared to the other models, but also with respect to prior results
            reported in these setups. See, e.g., the results reported in the original paper introducing this set of experiments (Hernández-Lobato & Adams, 2015), as well as numerous other BNN and GP papers on this setup who report variational inference results as part of their baselines (e.g., Bui et al., 2015; Wu et al., 2019; Haussmann et al., 2020,...).
    - A parallel question applies to VI on MNIST where it performs exceptionally bad, whereas I would expect a similar performance at least to the
        homoscedastic setup.
    - Wrt the runtime. VI should in practice be only slightly slower than the other methods. Especially given the
        statement in the appendix that for all methods in the non-image data, CPUs were sufficient, the argument on why
        the hyperparameter selection was limited is not completely clear to me. Can the authors elaborate?
    - Have the authors experimented with a simpler approach than Blundell et al.'s mixture prior, relying instead on a
        simple mean-field normal prior with local-reparameterization (Kingma et al., 2015) to stabilize the gradients?
        Then a single weight sample per forward pass should be sufficient to train the models without stability issues
        on both UCI as well as the image data sets without too many problems increasing the runtime cost to only twice
        as much as a deterministic net would cost.
- In UCI and CRISPR, the proposed MAP approach performed almost always similarly to/better than the Laplace approach.
    This behavior drastically changes for the synthetic image data experiments, especially FashionMNIST. Can the
    authors speculate/provide a discussion on why the additional structure provided provides this huge difference?



_____
Bui et al., 2016: Deep Gaussian Processes for Regression using Approximate Expectation Propagation
Haussmann et al., 2020: Sampling-Free Variational Inference of Bayesian Neural Networks by Variance Backpropagation
Kingma et al., 2015: Variational Dropout and the Local Reparameterization Trick
Wu et al., 2019: Deterministic Variational Inference for Robust Bayesian Neural Networks


**Limitations:**

Limitations of the model are discussed, but the societal impact is not. Given the theoretical nature of the paper,
I do not consider this lack of discussion to be a problem.

---

> ### Author Rebuttal · Authors · 2023-08-09
>
> Thank you for the overall positive evaluation of our paper and your detailed and constructive comments and questions!
>
> > Section 5.2 reads more like an afterthought that is added to the paper to have an additional experiment.
>
> Thank you for your comment. This is mostly due to lack of space. We will give the section more prominence, and elaborate more on its nature and relevance.
>
> > Evaluation of the results is vague. E.g., Heteroscedastic VI (PP) has a huge variance in Figure 1, Exp 1, whereas it suffers no such problems in the individual replications. The Laplace approach of the proposed method not only performs a little bit worse than the others in setting two (survival-screen-A375), but struggles a lot, whereas the MAP approach does not suffer from this problem.
>
> Regarding VI: during training on the mean response, some runs diverge to outlier values despite exhaustive tuning of hyperparameters and the use of common VI libraries and their recommended settings. Hence those lead to a large error (while the majority of runs achieves good results). We provide more details regarding the VI baseline below.
>
> Regarding Laplace: this is indeed curious and we will investigate further. We want to emphasize that these are challenging real-world datasets and we don't expect any single method to perform best on these. Despite that, we find it encouraging that our method performs best in two out of three settings.
>
> > I am not sure how to parse the sentence in l27/28. Hasn’t epistemic uncertainty modeling been the main focus for now with aleatoric uncertainty modeling being of secondary interest? Can the authors provide some intuition on what they mean here?
>
> In the case of heteroscedastic regression, recent approaches, e.g. the faithful and $\beta$-NLL methods, are (variations of) maximum likelihood estimation and therefore do not provide epistemic uncertainties. This observation has motivated our statement. On the other hand, the Bayesian community indeed focused more on epistemic uncertainties instead of aleatoric uncertainties. We will clarify this.
>
> > Figure 2: How does the corresponding test NLL look like for the proposed approach on this task?
>
> The NLL is reported in Table 2 on the left. Note that the homoscedastic baseline in Fig. 2 is different since we use the empirical Bayesian homoscedastic method in Table 2 for its better performance.
>
> > In UCI and CRISPR, the proposed MAP approach performed almost always similarly to/better than the Laplace approach. This behavior drastically changes for the synthetic image data experiments.
>
> We hypothesize that the proposed empirical Bayesian approach with Laplace particularly helps for deeper neural networks because it can adjust the prior precision/weight decay individually per layer, which is not tractable with cross-validation. On UCI and CRISPR, the networks only have one hidden layer. We therefore believe that the proposed benchmark is interesting for heteroscedastic neural network regression because it requires more complex networks than prior benchmarks.
>
> > Can the authors speculate on the performance of VI in the naval and plant experiments? In both data sets the performance is not just a drastic outlier compared to the other models, but also with respect to prior results reported in these setups. [...]. A parallel question applies to VI on MNIST where it performs exceptionally bad, whereas I would expect a similar performance at least to the homoscedastic setup.
>
> Thank you for drawing our attention to these matters. As stated above, we had training stability issues despite extensive manual tuning for VI and some runs end up diverging while others give good performance. As per your suggestion, we will improve our VI baselines with local reparameterization and simpler prior.
>
> > VI should in practice be only slightly slower than the other methods. Especially given the statement in the appendix that for all methods in the non-image data, CPUs were sufficient, the argument on why the hyperparameter selection was limited is not completely clear to me. Can the authors elaborate?
>
> VI runs 10 times slower due to 10 MC samples. Less MC samples during training showed even more convergence issues. Therefore, we tested hyperparameters manually to obtain a suitable grid for validation. Nonetheless, some outlier runs reduce the performance. As stated above, we hope to alleviate these issues with flipout or local reparameterization.
>
> > Have the authors experimented with a simpler approach than Blundell et al.'s mixture prior, relying instead on a simple mean-field normal prior with local-reparameterization (Kingma et al., 2015) to stabilize the gradients?
>
> Thank you for the suggestion. As stated above, we will experiment with these. However, we want to point out that we used a standard implementation for VI in neural networks (`blitz-bayesian-pytorch`) and followed suggested defaults (see also Appendix D.1.2 ll. 599-607). For now, we added the MFVI performance from the suggested DVI paper for comparison in Table 8 (see rebuttal PDF).
>
> > Minor comments and Typos
>
> Thanks for pointing these out. We will make the necessary changes.
>
> We will include the above clarifications, as well as the improvements mentioned in the general response into our manuscript. We hope that these improvements positively influence your assessment of our paper.

---

> > ### Comment · Reviewer_7fHN · 2023-08-16
> >
> > Thank you for the clarifications. I keep my score of recommending acceptance.
> >
> > Given the stability issues you report during training, I would, however, recommend to always rely on a local reparameterization from now on as it greatly reduces variance during training. I do not know the cited library, but you can find implementations in most major libraries, e.g., pyro for pytorch relies on it per default [1] and tensorflow probability [2] contains implementations of it as well. Inference without it just unnecessarily reduces (or even sometimes destroys as you observed) your numerical stability without there being a theoretical justification.
> >
> >
> > _____
> > [1] https://docs.pyro.ai/en/stable/contrib.bnn.html
> > [2] https://www.tensorflow.org/probability/api_docs/python/tfp/layers/DenseLocalReparameterization

---

> > > ### Author Response · Authors · 2023-08-21
> > >
> > > Thank you for your feedback. We greatly appreciate the advice! As also stated above, we will improve the VI baseline in the revised manuscript incorporating your suggestion.
> > >
> > > We already conducted some preliminary experiments, in which we observe that your recommended modifications improve the stability of the training of the VI baseline. As an example, see attached preliminary results for the CRISPR experiment, where we compare our previous VI implementation, `VI (Blundell)`, to `VI (Flipout + N prior)`, where we use Flipout and a simple Normal prior, in terms of test log-likelihood. Shown are mean and standard error over 10 seeds.
> > >
> > >
> > > |              | flow-cytometry-HEK293 | survival-screen-A375 | survival-screen-HEK293 |
> > > |--------------|-----------------------|----------------------|------------------------|
> > > | VI (Blundell)          | -1.30 (0.590)         | -0.26 (0.006)        | -0.15 (0.008)          |
> > > | VI (Flipout + N prior) | -0.40 (0.006)         | -0.27 (0.007)        | -0.26 (0.01)           |

---

> > > > ### Comment · Reviewer_7fHN · 2023-08-22
> > > >
> > > > Thanks for the update!

---

### Official Review · Reviewer_HwYi · 2023-07-06

**Soundness:** 2 fair
**Presentation:** 2 fair
**Contribution:** 3 good
**Rating:** 6
**Confidence:** 4

**Summary:**

This paper extends the technique of the Laplace Approximation to incorporate heteroscedastic aleatoric uncertainty. The proposed technique achieves this by exploiting the natural parameters of the Gaussian likelihood. Main motivation is the coupling of the mean and the input dependent variance in the Gaussian likelihood. Using the natural parameters, the paper suggests the heteroscedastic Gaussian log-likelihood, linearized Laplace Approximation, and marginal likelihood for obtaining the prior precision term. Two standard benchmarks including UCI and one self-made benchmarks, improvements over the previous methods are demonstrated.

**Strengths:**

In my opinion, the followings are the strengths of this paper:

- Presentation of the paper is generally clear to see the contributions of the paper. I appreciated especially the section 2, which carefully addresses the problem at hand.

- The contribution of the paper is relevant to the current Bayesian Deep Learning community. For example, extension of the Laplace Approximation to consider input dependent aleatoric uncertainty is meaningful. To do so, the paper provides long stretch from finding the problem in standard formulation of the Gaussian likelihood, to showing the adaptations on linearized Laplace Approximation, Gaussian log likelihood, predictive distribution and marginal log likelihood.

- Experimental results show good performance.

**Weaknesses:**

The followings might be the weaknesses of the paper (all major):

- The experiments only focus on regression problem, which is limited in scope for a paper on Bayesian Deep Learning.

The paper mainly focuses on the regression problems, e.g., all the experiments as well as the methodologies (section 2 and section 5). One way to address this point could be certain changes in the title. Would it be possible to include the term "regression" in the title?

- Certain parts of the paper need revision in presentation.

1. Many technical terms are introduced without explaining them well. Major examples are in the introduction 2nd paragraph, e.g., feasible generalized least squares, natural parameters, standard FGLS, etc.

2. The logic flow in the introduction may not be very kind to the reader. In particular, I had difficult time reading the 3rd paragraph. After reading the paper, I could grasp the concept well, but my feeling is that it is diving too much into the details early on.

3. I left minor comments later in the review.

- The baselines are restricted to the area of heteroscedastic aleatoric uncertainty estimation, which may be limited for broader audience. The scale of the experiments are also limited to toy-ish benchmarks. One option would be to use the uncertainty-baselines benchmarks.

In all the experiments, the baselines are rather restricted. While these choices validate the main point of the paper, I think the paper could improve by also competing against generic state of the art in uncertainty estimation. Would it be possible to include atleast MC-dropout, deep ensemble, and maybe stochastic HMC? (with homoscedastic aleatoric terms?) In this way, one could examine the importance of heteroscedastic aleatoric uncertainty estimation.

- Related work section is missing in the paper, which makes it harder to locate this work within the state-of-the-art.

To back up, there is no related work section in the main body of the paper. This makes it difficult to locate this work within the current state of the art. Would it be possible to include them? (one could shorten other parts of the paper). Some important areas are the usage of natural parameters of the Gaussian distribution, and more detailed treatment on aleatoric uncertainty, e.g., calibration methods, combination of both model and data uncertainty, etc. To name only few, highly relevant works seem to be: (1) Natural-Parameter Networks: A Class of Probabilistic Neural Networks (one of the works that uses natural parameters in more general context) and (2) estimating model uncertainty of neural networks in sparse information form (one of the works that uses natural parameters of Gaussian distribution for Laplace Approximation). Why the authors build on regularization based aleatoric uncertainty estimation method should also be mentioned.

**Questions:**

(minor comments/questions)

- in lines 20-21: isnt active learning and reinforcement learning part of the decision making problems?

- in lines 22-24: there are lots of work on making data noise as function of inputs. It might be an outdated statement.

- lines 48-54 should be made easier to read.

- Can you comment on the practical relevance of the problem 2.1?

- Is it possible to refer to all the derivations? (if they are present in the appendix?)

- line 198: is it a typo the jacobian term?

- is it possible to comment on the complexity of the overall pipeline, when compared to the standard linearized Laplace Approximation?

**Limitations:**

The limitations sections exist.

---

> ### Author Rebuttal · Authors · 2023-08-09
>
> Thank you for your detailed feedback and constructive questions which will help inprove our manuscript.
>
> > The paper mainly focuses on the regression problems. [...] Would it be possible to include the term "regression" in the title?
>
> Thank you for pointing out that heteroscedastic does not imply the focus on regression. We are happy to adjust the title to include "regression", for example, "Effective Bayesian Heteroscedastic Regression: Model Selection and Epistemic Uncertainties".
>
> > The baselines are restricted to the area of heteroscedastic aleatoric uncertainty estimation, which may be limited for broader audience. [...] Would it be possible to use the uncertainty-baselines benchmarks?
>
> We believe that the problem of regression, as opposed to classification, is similarly important in practice and for the broader audience but relatively underexplored in deep learning. In fact, the uncertainty-baselines benchmarks illustrate this issue since they exclusively include classification problems.
>
> > In all the experiments, the baselines are rather restricted. [...] Would it be possible to include atleast MC-dropout, deep ensemble, and maybe stochastic HMC with homoscedastic aleatoric terms?
>
> Thanks for these suggestions. We are happy to include these as baselines. For UCI regression datasets, we include the results provided by (Wu et al, 2019; Gal et al 2016). These works use the same setup as we do, so results are comparable and already provide evidence that our approach outperforms these additional baselines, speaking to the significance of our work. However, we want to note that deep ensembles are applicable to all the baselines and our method, see Eschenhagen et al. (2021). We would therefore add an ablation that studies the effect of additional ensembling for ours and other methods. We are happy to extend these baselines to all our experiments in the camera-ready version and will assess the feasibility of an HMC baseline.
>
> References:
>
> - Wu A., et al. Deterministic Variationa Infeference for Robust Bayesian Neural Networks. In ICLR, 2019.
> - Gal Y., and Ghahramani Z. Dropout as a Bayesian Approximation: Representing Model Uncertainty in Deep Learning. In ICML, 2016.
> - Eschenhagen R., et al. Mixtures of Laplace approximations for improved post-hoc uncertainty in deep learning. Bayesian Deep Learning Workshop 2021.
>
>
> > Certain parts of the paper need revision in presentation. [...] undefined technical terms, confusing introduction.
>
> Thank you for the clear description of what was unclear. We will improve the text accordingly.
>
> > Related work section is missing in the paper, which makes it harder to locate this work within the state-of-the-art. [...]
>
> We understand that while we discuss the most recent related work in great detail (Sec. 2, Appendix B and C), we should improve the discussion of related work in a broader context. Thus, we will extend the discussion of related work in Sec. 2, making sure that we cover the Baselines that we now added, and your suggestions discussed below.
>
> > relevant works seem to be: (1) Natural-Parameter Networks: A Class of Probabilistic Neural Networks and (2) estimating model uncertainty of neural networks in sparse information form.
>
> Thank you for the additional references, we will refer to them but want to point out that they are rather complementary to our work: (1) uses natural exponential families for posterior approximations in Bayesian neural networks and (2) could be used as alternative to KFAC for the Laplace posterior approximation. Both only apply to homoscedastic regression but could very well be combined with our approach to be applicable for heteroscedastic regression.
>
> > Isn't active learning and reinforcement learning part of the decision making problems?
>
> Thanks for pointing out this error, we will fix it.
>
> > Suggestions made in the "Questions" section
>
> We will make the corresponding changes and clarifications: 1) active and reinforcement learning part of decision making problems; 2) correct lines 22-24; 3) clarify lines 48-54; 4) refer to derivations in appendix; 5) clarify Jacobian term.
>
> > Can you comment on the practical relevance of the problem 2.1?
>
> Recent works on heteroscedastic regression focus on ad-hoc regularizations that seem to work for certain problems but fall short when applied to more complex problems. Problem 2.1 is a prototype of a complex heteroscedastic regression problem that a good algorithm should be able to solve.
>
> > Is it possible to comment on the complexity of the overall pipeline, when compared to the standard linearized Laplace Approximation?
>
> The standard linearized Laplace approximation is not applicable to the heteroscedastic case due to potentially negative definite Hessian, which our approach fixes using natural parameters. The complexity of our approximation is the same as for homoscedastic regression. This also shows in the empirical measurements in Table 9 of the rebuttal pdf where homoscedastic and heteroscedastic inference have the same runtime. We will further clarify the computational complexities in the appendix.
>
> We hope that we could address your questions and remarks in our response and the attached pdf with additional results. If so, we would appreciate if you considered revising your score.

---

> > ### Comment · Reviewer_HwYi · 2023-08-16
> > **On author response**
> >
> > I would like to thank the authors for the efforts involved. I have read the rebuttal as well as that of other reviewers.
> >
> > I decided to increase the score, given that the promises made during the rebuttal are properly address in the final revision.

---

### Official Review · Reviewer_PsPP · 2023-07-06

**Soundness:** 3 good
**Presentation:** 3 good
**Contribution:** 3 good
**Rating:** 7
**Confidence:** 3

**Summary:**

This article focuses on refining the techniques used to manage two types of uncertainties in complex regression tasks utilizing deep neural networks. The uncertainties, known as aleatoric (arising from inherent randomness in the data) and epistemic (originating from the model's limitations), are critical to address for robust and accurate predictions. The authors propose an innovative approach using the natural parameterization of the Gaussian likelihood to overcome the gradient scaling issue commonly encountered in traditional methodologies. Additionally, they introduce an efficient Laplace approximation that enhances heteroscedastic neural networks, providing epistemic uncertainties, and facilitating automatic regularization through empirical Bayes. This method outperforms earlier strategies in heteroscedastic regression, demonstrating scalability and obviating the need for hyperparameter tuning. Empirical validation on diverse datasets, including a new image dataset, UCI regression, and CRISPR-Cas13, yielded superior performance, suggesting potential applicability to other real-world datasets in the future.

**Strengths:**

- They are addressing very important shortcomings in one of the best algorithms for quantifying uncertainty using neural networks.
- Their method seems mathematically solid and is supported by extensive experiments.
- The code is available, and this work seems to be reproducible.
- Their method is thoroughly compared with the rival algorithms.
- Compared to rival methods, this method has less hyperparameters.



**Weaknesses:**

I cannot spot any weaknesses in this paper.

**Questions:**

- How large is the computation cost and time of the proposed method in comparison to the rival methods? In case it is significantly heavier, it should be properly discussed in the limitation section.

**Limitations:**

The limitations of this work are discussed properly.

---

> ### Author Rebuttal · Authors · 2023-08-09
>
> Thank you for your positive remarks! Below we would like to clarify your concerns regarding computational cost of our method.
>
> > How large is the computation cost and time of the proposed method in comparison to the rival methods?
>
> 1. Using the natural parameterization of the likelihood does not change the computational cost or complexity.
> 2. Using the Laplace posterior and predictive incurs a one-time cost of computing the approximation (Sec. 4.1) and the posterior predictive is slightly more expensive (Sec. 4.3) with $\mathcal{O}(P\sqrt{P})$ instead of $\mathcal{O}(P)$ for an MLP with fixed hidden layer sizes using KFAC. Empirically, the predictives are very fast and this is not a bottleneck. In comparison to the standard predictive, Table 9 in the rebuttal pdf shows that the posterior predictive is only 5x slower than a single forward pass without epistemic uncertainty.
> 3. Using empirical Bayes even **reduces** the computational cost because it only requires a single training run in comparison to validation-based selection of the regularization. This is apparent from Table 9 as well where the runtime of empirical Bayes is roughly 5 times faster when the hyperparameter tuning is included.
>
> Overall, the proposed natural Laplace method leads to faster training than rival methods when including the regularization strength selection and a slight increase in runtime of the posterior predictive.
>
> We will clarify the above points in Sec. 4.4 and hope that our answer alleviates your concerns regarding the computational cost and time of our method. If so, we would appreciate if you considered revising your score.

---

> > ### Comment · Reviewer_PsPP · 2023-08-14
> >
> > Thank you for the extra information provided. I will modify my assessment accordingly.

---

### Official Review · Reviewer_2cQi · 2023-07-07

**Soundness:** 4 excellent
**Presentation:** 4 excellent
**Contribution:** 3 good
**Rating:** 7
**Confidence:** 4

**Summary:**

This paper introduces a new method for fitting heteroscedastic regression models with neural networks. In contrast to previous work that highlights the deficiencies of MLE and indirectly hint at regularization, this paper suggests using natural gradients during training and combines Bayesian ideas for regularization and to handle uncertainty. This modeling approach accounts for both aleatoric and epistemic uncertainty through the posterior predictive distribution. Bayesian inference can be computationally expensive, so they utilize a Laplace approximation to the posterior and simplifying factorizations on the Hessian. Natural gradients avoid some of the issues with gradients that occur when training with other parameterizations ($\mathcal{N}(\mu, \sigma^2)$). Empirically this method achieves strong results on real-world data for regression tasks as well as on image data.

**Strengths:**

- Problem is well-motivated, and the distinctions between this method and existing ones are clear
- Experiments show solid results and extend the method to a setting with image data as opposed to only tabular data
- Combines ideas from approximate Bayesian inference with a parameterization commonly used for Gaussian processes

**Weaknesses:**

- typo line 98: "the mean respectively the covariance matrix" --> "the mean and covariance matrix, respectively"
- Structurally, it was odd to see Problem 2.1 presented so high up in the paper
- I would appreciate more discussion on the differences between the natural MAP vs natural Laplace methods. In particular, the differences in the interpretations of the two models (what sorts of uncertainties they account for) and when to use which

**Questions:**

- How does this method perform on data that is homoscedastic (ie the model is misspecified for the task)? Does the variance function flatten out, or will it overfit the variances (or $\eta_2$)?

- Does empirical Bayes put this method at risk of overfitting?

- Is $\eta_2(\cdot)$ necessary if it gets omitted when estimating the epistemic uncertainty (line 250) and what was the reason for this choice if it could have been done at low cost?

- How does this type of "Bayesian-ness" compare to the ideas presented in Stirn and Knowles (2020) or Detlefsen et al (2019) in terms of interpretation?



**Limitations:**

There was not much discussion of this, but that is appropriate for the nature of this work.

---

> ### Author Rebuttal · Authors · 2023-08-09
>
> Thank you for your positive evaluation of our work and your constructive comments and questions.
>
> **Weaknesses**
>
> > It was odd to see Problem 2.1 presented so high up in the paper.
>
> Currently, Section 2 serves as a theoretical comparison to the most related works. Within this context, we introduce the problem for the case-study presented in Figure 2 to highlight the conceptual difference to our approach. Given your feedback and the feedback of the other reviewers, we plan to make the discussion of related work more direct in this section complementing our discussion in the introduction, Section 2 and Appendix B and C. We can defer the definition to the experiments. We also welcome other suggestions.
>
> > I would appreciate more discussion on the differences between the natural MAP vs natural Laplace methods.
>
> These only differ in how the regularization is optimized: for "MAP", we optimize the regularization on a validation set, while for "Laplace", we use empirical Bayes (ML-II; see Sec. 4.2) to optimize the regularization jointly on the training data. We provide the pseudocode for the optimization in Appendix E1: for "MAP" the optimization of $\delta$ and the corresponding hyperparameter update are omitted. We will make this clearer by adding replacing Laplace with "empirical Bayes (EB)".
>
> **Questions**
>
> > How does this method perform on data that is homoscedastic (ie the model is misspecified for the task)? Does the variance function flatten out, or will it overfit the variances (or $\eta_2$)?
>
> Thanks for the interesting question. We added results for this on the Skafte and image regression tasks with modified homoscedastic noise to the rebuttal pdf in Figure 8. We indeed find that our method successfully regularizes towards a homoscedastic aleatoric uncertainty.
>
> > Does empirical Bayes put this method at risk of overfitting?
>
> Since we only optimize the regularization strength, the potential to overfit is quite limited. Our experiments confirm this: we do not find it to overfit compared to validation-based selection of the regularization.
>
> > Is $\eta_2$ necessary if it gets omitted when estimating the epistemic uncertainty (line 250) and what was the reason for this choice if it could have been done at low cost?
>
> Yes, $\eta_2$ is necessary since the Jacobian $J_\mu(x)$ depends on it and is required to approximate the epistemic uncertainty. This is a low-cost estimate since it is in a closed form.
>
> > How does this type of "Bayesian-ness" compare to the ideas presented in Stirn and Knowles (2020) or Detlefsen et al (2019) in terms of interpretation?
>
> Stirn & Knowles (2020) and Detlefsen et al. (2019) propose non-Bayesian methods and therefore infer only heteroscedastic aleatoric uncertainties $\sigma^2(x)$. They have no epistemic uncertainty, which would require uncertainty about the model or its parameters. For a visual depiction of the additional epistemic uncertainty, see "Heteroscedastic Laplace" on the right in Fig. 1.
>
> We hope that we could clarify your questions and if so, would appreaciate if you considered revising your score.

---

> > ### Comment · Reviewer_2cQi · 2023-08-17
> >
> > Thanks for the extra information and clarifications. I will update my score.

---

### Author Rebuttal · Authors · 2023-08-09

We thank the reviewers for their time and constructive feedback on our manuscript. It is encouraging to see that the reviewers rated our paper overall positively pointing out that our paper *"identifies prior weaknesses, and offers a principled solution"* (Reviewer 7fHN), that Reviewer HwYi *"appreciated especially the section 2, which carefully addresses the problem at hand"*, that our *"method seems mathematically solid and is supported by extensive experiments"* (Reviewer PsPP) and lastly that Reviewer 2cQi acknowledges our dedication to *"extend the method to a setting with image data as opposed to only tabular data"*.

Further, as asked by the reviewers, we investigated the following additional aspects (cf. attached pdf):

* **Baselines (HwYi, 7fHN)**: After confirming that the papers of Wu et al., (2019), and Gal and Ghahramani (2016) follow the same experimental setup as we do for the UCI datasets, we included their results for MFVI (Graves, 2011), DVI (Wu et al., 2019), and MC Dropout (Gal and Ghahramani, 2016) as shown in Table 8. We find that our proposed approaches still reach the overall best performance.
* **Homoscedastic Data (2cQi)**: We performed additional experiments to confirm that our approach also performs well on homoscedastic data: a) we plot the mean and uncertainty estimates for a homoscedastic variant of the data generated for Figure 1 and b) compare both on a homoscedastic variant of the image regression task in Table. The results are in Figure 8 of the rebuttal pdf. They show that the proposed approach does not overfit and instead recovers the homoscedastic aleatoric uncertainty.
* **Runtime (PsPP)** In general, we would like to note that the Laplace methods do not require cross-validation, which is necessary for MAP and the remaining baselines and therefore have an advantage in terms of runtime. In Table 9 of the rebuttal, we show runtimes for the image regression task for training (including hyperparameters) and inference.
* **Visualizations** In addition to showing the differences between MAP and Laplace, we also plot the mean and uncertainty estimates for MC Dropout and MFVI for the example provided in Figure 1 (cf. Figure 7, rebuttal pdf)---showing a favourable fit for our method.

Beyond these experiments, we will also add MC Dropout and an improved VI baseline to the remaining experiments, and add an ablation in which we instantiate our methods and baselines as deep ensembles (Reviewer HwYi). Further, we will extend Section 2 to discuss more related work---especially from the Bayesian learning literature as suggested by the reviewers, and lastly, include all clarifications provided during the rebuttal into our manuscript.

- Wu A., et al. Deterministic Variational Infeference for Robust Bayesian Neural Networks. In ICLR, 2019.
- Gal Y., and Ghahramani Z. Dropout as a Bayesian Approximation: Representing Model Uncertainty in Deep Learning. In ICML, 2016.
- Graves A. Practical variational inference for neural networks. In NeurIPS, 2011

---

### Decision · Program_Chairs · 2023-09-21

**Decision:**

Accept (poster)

**Comment:**

This paper addresses an important problem: inappropriate homoskedasticity assumptions in Bayesian neural nets (or any model, really) make it impossible for Bayes's rule to yield reasonable uncertainty estimates (and can also distort point estimates). The reviewers agree that this paper offers a principled, effective recipe for addressing this problem, and are all in favor of acceptance. I look forward to seeing the final version with the reviewers' suggestions incorporated.